# COMPARISONS ARE ALL YOU NEED FOR OPTIMIZING SMOOTH FUNCTIONS

## ABSTRACT

When optimizing machine learning models, there are various scenarios where gradient computations are challenging or even infeasible. Furthermore, in reinforcement learning (RL), preference-based RL that only compares between options has wide applications, including reinforcement learning with human feedback in large language models. In this paper, we systematically study optimization of a smooth function $f: \mathbb{R}^n \to \mathbb{R}$ only assuming an oracle that compares function values at two points and tells which is larger. When $f$ is convex, we give two algorithms using $\tilde{O}(n/\epsilon)$ and $\tilde{O}(n^2)$ comparison queries to find an $\epsilon$-optimal solution, respectively. When $f$ is nonconvex, our algorithm uses $\tilde{O}(n/\epsilon^2)$ comparison queries to find an $\epsilon$-approximate stationary point. All these results match the best-known zeroth-order algorithms with function evaluation queries in $n$ dependence, thus suggesting that *comparisons are all you need for optimizing smooth functions using derivative-free methods*. In addition, we also give an algorithm for escaping saddle points and reaching an $\epsilon$-second order stationary point of a nonconvex $f$, using $\tilde{O}(n^{1.5}/\epsilon^{2.5})$ comparison queries.

## 1 INTRODUCTION

Optimization is pivotal in the realm of machine learning. For instance, advancements in stochastic gradient descent (SGD) such as ADAM (Kingma & Ba, 2015), Adagrad (Duchi et al., 2011), etc., serve as foundational methods for the training of deep neural networks. However, there exist scenarios where gradient computations are challenging or even infeasible, such as black-box adversarial attack on neural networks (Papernot et al., 2017; Madry et al., 2018; Chen et al., 2017) and policy search in reinforcement learning (Salimans et al., 2017; Choromanski et al., 2018). Consequently, zeroth-order optimization methods with function evaluations have gained prominence, with provable guarantee for convex optimization (Duchi et al., 2015; Nesterov & Spokoiny, 2017) and nonconvex optimization (Ghadimi & Lan, 2013; Fang et al., 2018; Jin et al., 2018a; Ji et al., 2019; Zhang et al., 2022; Vlatakis-Gkaragkounis et al., 2019; Balasubramanian & Ghadimi, 2022).

Furthermore, optimization for machine learning has been recently soliciting for even less information. For instance, it is known that taking only signs of gradient descents still enjoy good performance (Liu et al., 2019; Li et al., 2023; Bernstein et al., 2018). Moreover, in the breakthrough of large language models (LLMs), reinforcement learning from human feedback (RLHF) played an important rule in training these LLMs, especially GPTs by OpenAI (Ouyang et al., 2022). Compared to standard RL that applies function evaluation for rewards, RLHF is preference-based RL that only compares between options and tells which is better. There is emerging research interest in preference-based RL, where various works have established provable guarantees for learning a near-optimal policy from preference feedback (Chen et al., 2022; Saha et al., 2023; Novoseller et al., 2020; Xu et al., 2020; Zhu et al., 2023; Tang et al., 2023). Furthermore, Wang et al. (2023) proved that for a wide range of preference models, preference-based RL can be solved with small or no extra costs compared to those of standard reward-based RL.

In this paper, we systematically study optimization of smooth functions using comparisons. Specifically, for a function $f: \mathbb{R}^n \to \mathbb{R}$, we define the *comparison oracle* of $f$ as $O_f^{\text{Comp}}: \mathbb{R}^n \times \mathbb{R}^n \to$

$\{-1, 1\}$ such that

$$O_f^{\text{Comp}}(\mathbf{x}, \mathbf{y}) = \begin{cases} 1 & \text{if } f(\mathbf{x}) \geq f(\mathbf{y}) \\ -1 & \text{if } f(\mathbf{x}) \leq f(\mathbf{y}) \end{cases}. \tag{1}$$

(When $f(\mathbf{x}) = f(\mathbf{y})$, outputting either 1 or $-1$ is okay.) We consider an $L$-smooth function $f \colon \mathbb{R}^n \to \mathbb{R}$, defined as

$$\|\nabla f(\mathbf{x}) - \nabla f(\mathbf{y})\| \leq L\|\mathbf{x} - \mathbf{y}\| \quad \forall \mathbf{x}, \mathbf{y} \in \mathbb{R}^n.$$

Furthermore, we say $f$ is $\rho$-*Hessian Lipschitz* if

$$\|\nabla^2 f(\mathbf{x}) - \nabla^2 f(\mathbf{y})\| \leq \rho\|\mathbf{x} - \mathbf{y}\| \quad \forall \mathbf{x}, \mathbf{y} \in \mathbb{R}^n.$$

In terms of the goal of optimization, we define:

- $\mathbf{x} \in \mathbb{R}^n$ is an $\epsilon$-*optimal point* if $f(\mathbf{x}) \leq f^* + \epsilon$, where $f^* := \inf_{\mathbf{x}} f(\mathbf{x})$.
- $\mathbf{x} \in \mathbb{R}^n$ is an $\epsilon$-*first-order stationary point ($\epsilon$-FOSP)* if $\|\nabla f(\mathbf{x})\| \leq \epsilon$.
- $\mathbf{x} \in \mathbb{R}^n$ is an $\epsilon$-*second-order stationary point ($\epsilon$-SOSP)* if $\|\nabla f(\mathbf{x})\| \leq \epsilon$ and $\lambda_{\min}(\nabla^2 f(\mathbf{x})) \geq -\sqrt{\rho\epsilon}$.[1]

**Our main results** can be listed as follows:

- For an $L$-smooth convex $f$, Theorem 2 finds an $\epsilon$-optimal point in $O(nL/\epsilon \log(nL/\epsilon))$ comparisons.
- For an $L$-smooth convex $f$, Theorem 3 finds an $\epsilon$-optimal point in $O(n^2 \log(nL/\epsilon))$ comparisons.
- For an $L$-smooth $f$, Theorem 4 finds an $\epsilon$-FOSP using $O(Ln \log n/\epsilon^2)$ comparisons.
- For an $L$-smooth, $\rho$-Hessian Lipschitz $f$, Theorem 5 finds an $\epsilon$-SOSP in $\tilde{O}(n^{1.5}/\epsilon^{2.5})$ comparisons.

Intuitively, our results can be described as **comparisons are all you need for derivative-free methods**: For finding an approximate minimum of a convex function, the state-of-the-art zeroth-order methods with full function evaluations have query complexities $O(n/\sqrt{\epsilon})$ (Nesterov & Spokoiny, 2017) or $\tilde{O}(n^2)$ (Lee et al., 2018), which are matched in $n$ by our Theorem 2 and Theorem 3 using comparisons, respectively. For finding an approximate stationary point of a nonconvex function, the state-of-the-art zeroth-order result has query complexity $O(n/\epsilon^2)$ (Fang et al., 2018), which is matched by our Theorem 4 up to a logarithmic factor. In other words, in derivative-free scenarios for optimizing smooth functions, function values per se are unimportant but their comparisons, which indicate the direction that the function decreases.

Among the literature for derivative-free optimization methods (Larson et al., 2019), direct search methods by Kolda et al. (2003) proceed by comparing function values, including the directional direct search method (Audet & Dennis Jr, 2006) and the Nelder-Mead method (Nelder & Mead, 1965) as examples. However, the directional direct search method does not have a known rate of convergence, meanwhile the Nelson-Mead method may fail to converge to a stationary point for smooth functions (Dennis & Torczon, 1991). As far as we know, the most relevant result is by Bergou et al. (2020), which proposed the stochastic three points (STP) method and found an $\epsilon$-optimal point of a convex function and an $\epsilon$-FOSP of a nonconvex function in $\tilde{O}(n/\epsilon)$ and $\tilde{O}(n/\epsilon^2)$ comparisons, respectively. STP also has a version with momentum (Gorbunov et al., 2020). Our Theorem 2 and Theorem 4 can be seen as rediscoveries of these results using different methods. In addition, literature on dueling convex optimization also achieves $\tilde{O}(n/\epsilon)$ for finding an $\epsilon$-optimal point of a convex function (Saha et al., 2021; 2022). However, for comparison-based convex optimization with $\text{poly}(\log 1/\epsilon)$ dependence, Jamieson et al. (2012) achieved this for strongly convex functions, and the state-of-the-art result for general convex optimization by Karabag et al. (2021) takes $\tilde{O}(n^4)$ comparison queries. Their algorithm applies the ellipsoid method, which has $\tilde{O}(n^2)$ iterations and each iteration takes $\tilde{O}(n^2)$ comparisons to construct the ellipsoid. This $\tilde{O}(n^4)$ bound is noticeably worse than our Theorem 3. As far as we know, our Theorem 5 is the *first provable guarantee* for finding an $\epsilon$-SOSP of a nonconvex function by comparisons.

---

[1]This is a standard definition among nonconvex optimization literature for escaping saddle points and reaching approximate second-order stationary points, see for instance (Nesterov & Polyak, 2006; Curtis et al., 2017; Agarwal et al., 2017; Carmon et al., 2018; Jin et al., 2018b; Allen-Zhu & Li, 2018; Xu et al., 2018; Zhang et al., 2022; Zhang & Gu, 2023).

**Techniques.** Our first technical contribution is Theorem 1, which for a point $\mathbf{x}$ estimates the direction of $\nabla f(\mathbf{x})$ within precision $\delta$. This is achieved by Algorithm 2, named as `Comparison-GDE` (GDE is the acronym for gradient direction estimation). It is built upon a directional preference subroutine (Algorithm 1), which inputs a unit vector $\mathbf{v} \in \mathbb{R}^n$ and a precision parameter $\Delta > 0$, and outputs whether $\langle \boldsymbol{\nabla} f(\mathbf{x}), \mathbf{v} \rangle \geq -\Delta$ or $\langle \boldsymbol{\nabla} f(\mathbf{x}), \mathbf{v} \rangle \leq \Delta$ using the value of the comparison oracle for $O_f^{\mathrm{Comp}}(\mathbf{x} + \frac{2\Delta}{L}\mathbf{v}, \mathbf{x})$. `Comparison-GDE` then has three phases:

- First, it sets $\mathbf{v}$ to be all standard basis directions $\mathbf{e}_i$ to determine the signs of all $\nabla_i f(\mathbf{x})$ (up to $\Delta$).
- It then sets $\mathbf{v}$ as $\frac{1}{\sqrt{2}}(\mathbf{e}_i - \mathbf{e}_j)$, which can determine whether $|\nabla_i f(\mathbf{x})|$ or $|\nabla_j f(\mathbf{x})|$ is larger (up to $\Delta$). Start with $\mathbf{e}_1$ and $\mathbf{e}_2$ and keep iterating to find the $i^*$ with the largest $|\frac{\partial}{\partial i^*} \nabla f(\mathbf{x})|$ (up to $\Delta$).
- Finally, for each $i \neq i^*$, It then sets $\mathbf{v}$ to have form $\frac{1}{\sqrt{1+\alpha_i^2}}(\alpha_i \mathbf{e}_{i^*} - \mathbf{e}_i)$ and applies binary search to find the value for $\alpha_i$ such that $\alpha_i |\nabla_{i^*} f(\mathbf{x})|$ equals to $|\nabla_i f(\mathbf{x})|$ up to enough precision.

`Comparison-GDE` outputs $\boldsymbol{\alpha}/\|\boldsymbol{\alpha}\|$ for GDE, where $\boldsymbol{\alpha} = (\alpha_1, \ldots, \alpha_n)^\top$. It in total uses $O(n \log(n/\delta))$ comparison queries, with the main cost coming from binary searches in the last step (the first two steps both take $\leq n$ comparisons).

We then leverage `Comparison-GDE` for solving various optimization problems. In convex optimization, we develop two algorithms that find an $\epsilon$-optimal point separately in Section 3.1 and Section 3.2. Our first algorithm is a specialization of the adaptive version of normalized gradient descent (NGD) introduced in Levy (2017), where we replace the normalized gradient query in their algorithm by `Comparison-GDE`. It is a natural choice to apply gradient estimation to normalized gradient descent, given that the comparison model only allows us to estimate the gradient direction without providing information about its norm. Note that Bergou et al. (2020) also discussed NGD, but their algorithm using NGD still needs the full gradient and cannot be directly implemented by comparisons. Our second algorithm builds upon the framework of cutting plane methods, where we show that the output of `Comparison-GDE` is a valid separation oracle, as long as it is accurate enough. Moreover, we note that Cai et al. (2022) also studied gradient estimation by comparisons and combined that with inexact NGD, but their complexity $\tilde{O}(d/\epsilon^{1.5})$ is suboptimal compared to ours.

In nonconvex optimization, we develop two algorithms that find an $\epsilon$-FOSP and an $\epsilon$-SOSP, respectively, in Section 4.1 and Section 4.2. Our algorithm for finding an $\epsilon$-FOSP is a specialization of the NGD algorithm, where the normalized gradient is given by `Comparison-GDE`. Our algorithm for finding an $\epsilon$-SOSP uses a similar approach as corresponding first-order methods by Allen-Zhu & Li (2018); Xu et al. (2018) and proceeds in rounds, where we alternately apply NGD and negative curvature descent to ensure that the function value will have a large decrease if more than $1/9$ of the iterations in this round are not $\epsilon$-SOSP. The normalized gradient descent part is essentially the same as our algorithm for $\epsilon$-FOSP in Section 4.1. The negative curvature descent part with comparison information, however, is much more technically involved. In particular, previous first-order methods (Allen-Zhu & Li, 2018; Xu et al., 2018; Zhang & Li, 2021) all contains a subroutine that can find a negative curvature direction near a saddle point $\mathbf{x}$ with $\lambda_{\min}(\nabla^2 f(\mathbf{x}) \leq -\sqrt{\rho\epsilon})$. One crucial step in this subroutine is to approximate the Hessian-vector product $\nabla^2 f(\mathbf{x}) \cdot \mathbf{y}$ for some unit vector $\mathbf{y} \in \mathbb{R}^n$ by taking the difference between $\nabla f(\mathbf{x} + r\mathbf{y})$ and $\nabla f(\mathbf{x})$, where $r$ is a very small parameter. However, this is infeasible in the comparison model which only allows us to estimate the gradient direction without providing information about its norm. Instead, we find the directions of $\nabla f(\mathbf{x}), \nabla f(\mathbf{x} + r\mathbf{y})$, and $\nabla f(\mathbf{x} - r\mathbf{y})$ by `Comparison-GDE`, and we determine the direction of $\nabla f(\mathbf{x} + r\mathbf{y}) - f(\mathbf{y})$ using the fact that its intersection with $\nabla f(\mathbf{x})$ and $\nabla f(\mathbf{x} + r\mathbf{y})$ as well as its intersection with $\nabla f(\mathbf{x})$ and $\nabla f(\mathbf{x} - r\mathbf{y})$ give two segments of same length (see Figure 1).

**Open questions.** Our work leaves several natural directions for future investigation:

- Can we give comparison-based optimization algorithms based on accelerated gradient descent (AGD) methods? This is challenging because AGD requires carefully chosen step sizes, but with comparisons we can only learn gradient directions but not the norm of gradients. This is also the main reason why the $1/\epsilon$ dependence in our Theorem 2 and Theorem 5 are worse than Nesterov & Spokoiny (2017) and Zhang & Gu (2023) with evaluations in their respective settings.

Figure 1: The intuition of Algorithm 10 for computing Hessian-vector products using gradient directions.

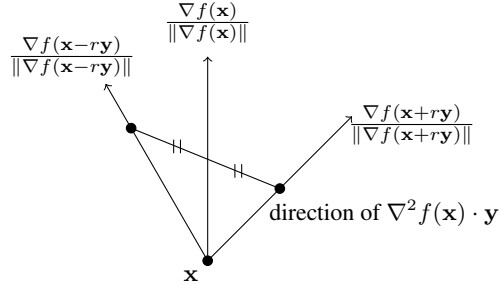

- Can we improve our result for finding second-order stationary points in nonconvex optimization? Compared to gradient-based methods that choose the step size in negative curvature finding (Allen-Zhu & Li, 2018; Xu et al., 2018), our comparison-based perturbed normalized gradient descent (Algorithm 5) can only utilize gradient directions but have no information about gradient norms, resulting in a fixed and conservative step size and in total $\tilde{O}(\sqrt{n}/\epsilon)$ iterations.

- Can we apply our algorithms to machine learning? Tang et al. (2023) made attempts on preference-based RL, and it is worth further exploring whether we can prove more theoretical results for preference-based RL and other machine learning settings. It would be also of general interest to see if our results can provide theoretical justification for quantization in neural networks (Gholami et al., 2022).

**Notations.** We use bold letters, e.g., $\mathbf{x}$, $\mathbf{y}$, to denote vectors and capital letters, e.g., $A$, $B$, to denote matrices. We use $\|\cdot\|$ to denote the Euclidean norm ($\ell_2$-norm) and denote $\mathcal{S}^{n-1}$ to be the $n$-dimensional sphere with radius 1, i.e., $\mathcal{S}^{n-1} := \{\mathbf{x} \in \mathbb{R}^n : \|\mathbf{x}\| = 1\}$. We denote $\mathbb{B}_R(\mathbf{x}) := \{\mathbf{y} \in \mathbb{R}^n : \|\mathbf{y} - \mathbf{x}\| \leq R\}$ and $[T] := \{0, 1, \ldots, T\}$. For a convex set $\mathcal{K} \subseteq \mathbb{R}^n$, its diameter is defined as $D := \sup_{\mathbf{x},\mathbf{y}\in\mathcal{K}} \|\mathbf{x} - \mathbf{y}\|$ and its projection operator $\Pi_{\mathcal{K}}$ is defined as

$$\Pi_{\mathcal{K}}(\mathbf{x}) := \operatorname{argmin}_{\mathbf{y}\in\mathcal{K}} \|\mathbf{x} - \mathbf{y}\|, \quad \forall \mathbf{x} \in \mathbb{R}^n.$$

## 2 ESTIMATION OF GRADIENT DIRECTION BY COMPARISONS

First, we show that given a point $\mathbf{x} \in \mathbb{R}^n$ and a direction $\mathbf{v} \in \mathbb{R}^n$, we can use one comparison query to understand whether the inner product $\langle \boldsymbol{\nabla} f(\mathbf{x}), \mathbf{v} \rangle$ is roughly positive or negative. Intuitively, this inner product determines whether $\mathbf{x} + \mathbf{v}$ is following or against the direction of $\boldsymbol{\nabla} f(\mathbf{x})$, also known as *directional preference* (DP) in Karabag et al. (2021).

**Lemma 1.** *Given a point $\mathbf{x} \in \mathbb{R}^n$, a unit vector $\mathbf{v} \in \mathbb{B}_1(0)$, and precision $\Delta > 0$ for directional preference. Then Algorithm 1 is correct:*

- *If $O_f^{\mathrm{Comp}}(\mathbf{x} + \frac{2\Delta}{L}\mathbf{v}, \mathbf{x}) = 1$, then $\langle \boldsymbol{\nabla} f(\mathbf{x}), \mathbf{v} \rangle \geq -\Delta$.*

- *If $O_f^{\mathrm{Comp}}(\mathbf{x} + \frac{2\Delta}{L}\mathbf{v}, \mathbf{x}) = -1$, then $\langle \boldsymbol{\nabla} f(\mathbf{x}), \mathbf{v} \rangle \leq \Delta$.*

---

**Algorithm 1:** DP($\mathbf{x}$,$\mathbf{v}$,$\Delta$)

---

**Input:** Comparison oracle $O_f^{\mathrm{Comp}}$ of $f \colon \mathbb{R}^n \to \mathbb{R}$, $\mathbf{x} \in \mathbb{R}^n$, unit vector $\mathbf{v} \in \mathbb{B}_1(0)$, $\Delta > 0$

1 **if** $O_f^{\mathrm{Comp}}(\mathbf{x} + \frac{2\Delta}{L}\mathbf{v}, \mathbf{x}) = 1$ **then**
2     **return** "$\langle \boldsymbol{\nabla} f(\mathbf{x}), \mathbf{v} \rangle \geq -\Delta$"
3 **else** (in this case $O_f^{\mathrm{Comp}}(\mathbf{x} + \frac{2\Delta}{L}\mathbf{v}, \mathbf{x}) = -1$)
4     **return** "$\langle \boldsymbol{\nabla} f(\mathbf{x}), \mathbf{v} \rangle \leq \Delta$"

---

*Proof.* Since $f$ is an $L$-smooth differentiable function,

$$|f(\mathbf{y}) - f(\mathbf{x}) - \langle \boldsymbol{\nabla} f(\mathbf{x}), \mathbf{y} - \mathbf{x} \rangle| \leq \frac{1}{2} L \|\mathbf{y} - \mathbf{x}\|^2$$

for any $\mathbf{x}, \mathbf{y} \in \mathbb{R}^n$. Take $\mathbf{y} = \mathbf{x} + \frac{2\Delta}{L} \mathbf{v}$, this gives

$$\left| f(\mathbf{y}) - f(\mathbf{x}) - \frac{2\Delta}{L} \langle \boldsymbol{\nabla} f(\mathbf{x}), \mathbf{v} \rangle \right| \leq \frac{1}{2} L \left( \frac{2\Delta}{L} \right)^2 = \frac{2\Delta^2}{L}.$$

Therefore, if $O_f^{\text{Comp}}(\mathbf{y}, \mathbf{x}) = 1$, i.e., $f(\mathbf{y}) \geq f(\mathbf{x})$,

$$\frac{2\Delta}{L} \langle \boldsymbol{\nabla} f(\mathbf{x}), \mathbf{v} \rangle \geq \frac{2\Delta}{L} \langle \boldsymbol{\nabla} f(\mathbf{x}), \mathbf{v} \rangle + f(\mathbf{x}) - f(\mathbf{y}) \geq -\frac{2\Delta^2}{L}$$

and hence $\langle \boldsymbol{\nabla} f(\mathbf{x}), \mathbf{v} \rangle \geq -\Delta$. On the other hand, if $O_f^{\text{Comp}}(\mathbf{y}, \mathbf{x}) = -1$, i.e., $f(\mathbf{y}) \leq f(\mathbf{x})$,

$$\frac{2\Delta}{L} \langle \boldsymbol{\nabla} f(\mathbf{x}), \mathbf{v} \rangle \leq f(\mathbf{y}) - f(\mathbf{x}) + \frac{2\Delta^2}{L} \leq \frac{2\Delta^2}{L}$$

and hence $\langle \boldsymbol{\nabla} f(\mathbf{x}), \mathbf{v} \rangle \leq \Delta$. $\qquad\square$

Now, we prove that we can use $\tilde{O}(n)$ comparison queries to approximate the direction of the gradient at a point, which is one of our main technical contributions.

**Theorem 1.** *For an $L$-smooth function $f \colon \mathbb{R}^n \to \mathbb{R}$ and a point $\mathbf{x} \in \mathbb{R}^n$, Algorithm 2 outputs an estimate $\tilde{\mathbf{g}}(\mathbf{x})$ of the direction of $\nabla f(\mathbf{x})$ using $O(n \log(n/\delta))$ queries to the comparison oracle $O_f^{\text{Comp}}$ of $f$ (Eq. (1)) that satisfies*

$$\left\| \tilde{\mathbf{g}}(\mathbf{x}) - \frac{\nabla f(\mathbf{x})}{\|\nabla f(\mathbf{x})\|} \right\| \leq \delta$$

*if we are given a parameter $\gamma > 0$ such that $\|\nabla f(\mathbf{x})\| \geq \gamma$.*

*Proof.* The correctness of (2) and (3) follows directly from the arguments in Line 2 and Line 3, respectively. For Line 6, since $\alpha_i \leq 1$ for any $i \in [n]$, the binary search can be regarded as having bins with interval lengths $\sqrt{1 + \alpha_i^2} \Delta \leq \sqrt{2} \Delta$, and when the binary search ends Eq. (4) is satisfied. Furthermore, Eq. (4) can be written as

$$\left| \alpha_i - \frac{g_i}{g_{i^*}} \right| \leq \frac{\sqrt{2}\Delta}{g_{i^*}} \leq \frac{2\Delta\sqrt{n}}{\gamma}.$$

This is because $\|\nabla f(\mathbf{x})\| = \|(g_1, \ldots, g_n)^\top\| \geq \gamma$ implies $\max_{i \in [n]} g_i \geq \gamma/\sqrt{n}$, and together with (3) we have $g_{i^*} \geq \gamma/\sqrt{n} - \sqrt{2}\Delta \geq \gamma/\sqrt{2n}$ because $\Delta \leq \gamma/4\sqrt{n}$.

We now estimate $\left\| \tilde{\mathbf{g}}(\mathbf{x}) - \frac{\nabla f(\mathbf{x})}{\|\nabla f(\mathbf{x})\|} \right\|$. Note $\frac{\nabla f(\mathbf{x})}{\|\nabla f(\mathbf{x})\|} = \frac{\nabla f(\mathbf{x})/g_{i^*}}{\|\nabla f(\mathbf{x})/g_{i^*}\|}$ and $\tilde{\mathbf{g}}(\mathbf{x}) = \boldsymbol{\alpha}/\|\boldsymbol{\alpha}\|$. Moreover

$$\left\| \boldsymbol{\alpha} - \frac{\nabla f(\mathbf{x})}{g_{i^*}} \right\| \leq \sum_{i=1}^n \left| \alpha_i - \frac{g_i}{g_{i^*}} \right| \leq \frac{2\Delta\sqrt{n}(n-1)}{\gamma}.$$

By Lemma 5 for bounding distance between normalized vectors) and the fact that $\|\boldsymbol{\alpha}\| \geq 1$,

$$\left\| \tilde{\mathbf{g}}(\mathbf{x}) - \frac{\nabla f(\mathbf{x})}{\|\nabla f(\mathbf{x})\|} \right\| = \left\| \frac{\boldsymbol{\alpha}}{\|\boldsymbol{\alpha}\|} - \frac{\nabla f(\mathbf{x})/g_{i^*}}{\|\nabla f(\mathbf{x})/g_{i^*}\|} \right\| \leq \frac{4\Delta n^{3/2}}{\gamma} \leq \delta.$$

Thus the correctness has been established. For the query complexity, Line 2 takes $n$ queries, Line 3 takes $n - 1$ queries, and Line 6 throughout the for loop takes $(n - 1)\lceil \log_2(\gamma/\sqrt{2}\Delta) + 1 \rceil = O(n \log(n/\delta))$ queries to the comparison oracle, given that each $\alpha_i$ is within the range of $[0, 1]$ and we approximate it to accuracy $\sqrt{2}\Delta/g_{i^*} \geq \sqrt{2}\Delta/\gamma$. This finishes the proof. $\qquad\square$

---

**Algorithm 2:** Comparison-based Gradient Direction Estimation (Comparison-GDE($\mathbf{x}, \delta, \gamma$))

**Input:** Comparison oracle $O_f^{\mathrm{Comp}}$ of $f \colon \mathbb{R}^n \to \mathbb{R}$, precision $\delta$, lower bound $\gamma$ on $\|\boldsymbol{\nabla} f(\mathbf{x})\|$

**1** Set $\Delta \leftarrow \delta\gamma/4n^{3/2}$. Denote $\boldsymbol{\nabla} f(\mathbf{x}) = (g_1, \ldots, g_n)^\top$

**2** Call Algorithm 1 with inputs $(\mathbf{x}, \mathbf{e}_1, \Delta), \ldots, (\mathbf{x}, \mathbf{e}_n, \Delta)$ where $e_i$ is the $i^{\text{th}}$ standard basis with $i^{\text{th}}$ coordinate being 1 and others being 0. This determines whether $g_i \geq -\Delta$ or $g_i \leq \Delta$ for each $i \in [n]$. WLOG

$$g_i \geq -\Delta \quad \forall i \in [n] \tag{2}$$

(otherwise take a minus sign for the $i^{\text{th}}$ coordinate)

**3** We next find the approximate largest one among $g_1, \ldots, g_n$. Call Algorithm 1 with input $(\mathbf{x}, \frac{1}{\sqrt{2}}(\mathbf{e}_1 - \mathbf{e}_2), \Delta)$. This determines whether $g_1 \geq g_2 - \sqrt{2}\Delta$ or $g_2 \geq g_1 - \sqrt{2}\Delta$. If the former, call Algorithm 1 with input $(\mathbf{x}, \frac{1}{\sqrt{2}}(\mathbf{e}_1 - \mathbf{e}_3), \Delta)$. If the later, call Algorithm 1 with input $(\mathbf{x}, \frac{1}{\sqrt{2}}(\mathbf{e}_2 - \mathbf{e}_3), \Delta)$. Iterate this until $e_n$, we find the $i^* \in [n]$ such that

$$g_{i^*} \geq \max_{i \in [n]} g_i - \sqrt{2}\Delta \tag{3}$$

**4 for** $i = 1$ to $i = n$ (except $i = i^*$) **do**

**5**      Initialize $\alpha_i \leftarrow 1/2$

**6**      Apply binary search to $\alpha_i$ in $\lceil \log_2(\gamma/\Delta) + 1 \rceil$ iterations by calling Algorithm 1 with input $(\mathbf{x}, \frac{1}{\sqrt{1+\alpha_i^2}}(\alpha_i \mathbf{e}_{i^*} - \mathbf{e}_i), \Delta)$. For the first iteration with $\alpha_i = 1/2$, if $\alpha_i g_{i^*} - g_i \geq -\sqrt{2}\Delta$ we then take $\alpha_i = 3/4$; if $\alpha_i g_{i^*} - g_i \leq \sqrt{2}\Delta$ we then take $\alpha_i = 1/4$. Later iterations are similar. Upon finishing the binary search, $\alpha_i$ satisfies

$$g_i - \sqrt{2}\Delta \leq \alpha_i g_{i^*} \leq g_i + \sqrt{2}\Delta \tag{4}$$

**7 return** $\tilde{\mathbf{g}}(\mathbf{x}) = \frac{\boldsymbol{\alpha}}{\|\boldsymbol{\alpha}\|}$ where $\alpha = (\alpha_1, \ldots, \alpha_n)^\top$, $\alpha_i$ ($i \neq i^*$) is the output of the for loop, $\alpha_{i^*} = 1$

---

## 3   Convex Optimization by Comparisons

In this section, we study convex optimization with function value comparisons:

**Problem 1** (Comparison-based convex optimization). *In the comparison-based convex optimization (CCO) problem we are given query access to a comparison oracle $O_f^{\mathrm{Comp}}$ (1) for an L-smooth convex function $f \colon \mathbb{R}^n \to \mathbb{R}$ whose minimum is achieved at $\mathbf{x}^*$ with $\|\mathbf{x}^*\| \leq R$. The goal is to output a point $\tilde{\mathbf{x}}$ such that $\|\tilde{\mathbf{x}}\| \leq R$ and $f(\tilde{\mathbf{x}}) - f(\mathbf{x}^*) \leq \epsilon$, i.e., $\tilde{\mathbf{x}}$ is an $\epsilon$-optimal point.*

We provide two algorithms that solve Problem 1. In Section 3.1, we use normalized gradient descent to achieve linear dependence in $n$ (up to a log factor) in terms of comparison queries. In Section 3.2, we use cutting plane method to achieve $\log(1/\epsilon)$ dependence in terms of comparison queries.

### 3.1   Comparison-based adaptive normalized gradient descent

In this subsection, we present our first algorithm for Problem 1, Algorithm 3, which applies `Comparison-GDE` (Algorithm 2) with estimated gradient direction at each iteration to the adaptive normalized gradient descent (AdaNGD), originally introduced by Levy (2017).

**Theorem 2.** *Algorithm 3 solves Problem 1 using $O(nLR^2/\epsilon \log(nLR^2/\epsilon))$ queries.*

The following result bounds the rate at which Algorithm 3 decreases the function value of $f$.

**Lemma 2.** *In the setting of Problem 1, Algorithm 3 satisfies*

$$\min_{t \in [T]} f(\mathbf{x}_t) - f^* \leq 2L(2R\sqrt{2T} + 2T\delta R)^2/T^2,$$

---

**Algorithm 3:** Comparison-based Approximate Adaptive Normalized Gradient Descent (Comparison-AdaNGD)

---

**Input:** Function $f \colon \mathbb{R}^n \to \mathbb{R}$, precision $\epsilon$, radius $R$

1   $T \leftarrow \frac{64LR^2}{\epsilon}, \delta \leftarrow \frac{1}{4R}\sqrt{\frac{\epsilon}{2L}}, \gamma \leftarrow \frac{\epsilon}{2R}, \mathbf{x}_0 \leftarrow \mathbf{0}$

2   **for** $t = 0, \dots, T-1$ **do**

3      $\hat{\mathbf{g}}_t \leftarrow \texttt{Comparison-GDE}(\mathbf{x}_t, \delta, \gamma)$

4      $\eta_t \leftarrow R\sqrt{2/t}$

5      $\mathbf{x}_{t+1} = \Pi_{\mathbb{B}_R(\mathbf{0})}(\mathbf{x}_t - \eta_t \hat{\mathbf{g}}_t)$

6   $t_{\text{out}} \leftarrow \operatorname{argmin}_{t \in [T]} f(\mathbf{x}_t)$

7   **return** $\mathbf{x}_{t_{\text{out}}}$

---

*if at each step we have*

$$\left\| \tilde{\mathbf{g}}_t - \frac{\nabla f_t(\mathbf{x}_t)}{\|\nabla f_t(\mathbf{x}_t)\|} \right\| \leq \delta \leq 1.$$

The proof of Lemma 2 is deferred to Appendix B. We now prove Theorem 2 using Lemma 2.

*Proof of Theorem 2.* We show that Algorithm 3 solves Problem 1 by contradiction. Assume that the output of Algorithm 3 is not an $\epsilon$-optimal point of $f$, or equivalently, $f(\mathbf{x}_t) - f^* \geq \epsilon$ for any $t \in [T]$. This leads to

$$\|\nabla f(\mathbf{x}_t)\| \geq \frac{f(\mathbf{x}_t) - f^*}{\|\mathbf{x}_t - \mathbf{x}^*\|} \geq \frac{\epsilon}{2R}, \quad \forall t \in [T]$$

given that $f$ is convex. Hence, Theorem 1 promises that

$$\left\| \hat{\mathbf{g}}_t - \frac{\nabla f(\mathbf{x}_t)}{\|\nabla f(\mathbf{x}_t)\|} \right\| \leq \delta \leq 1.$$

With these approximate gradient directions, by Lemma 2 we can derive that

$$\min_{t \in [T]} f(\mathbf{x}_t) - f^* \leq 2L(2R\sqrt{2T} + 2T\delta R)^2/T^2 \leq \epsilon,$$

contradiction. This proves the correctness of Algorithm 3. The query complexity of Algorithm 3 only comes from the gradient direction estimation step in Line 3, which equals

$$T \cdot O(n \log(n/\delta)) = O\left( \frac{nLR^2}{\epsilon} \log\left( \frac{nLR^2}{\epsilon} \right) \right).$$

$\square$

## 3.2   COMPARISON-BASED CUTTING PLANE METHOD

In this subsection, we provide a comparison-based cutting plane method that solves Problem 1. We begin by introducing the basic notation and concepts of cutting plane methods, which are algorithms that solves the feasibility problem defined as follows.

**Problem 2** (Feasibility Problem, Jiang et al. (2020); Sidford & Zhang (2023))**.** *We are given query access to a separation oracle for a set $K \subset \mathbb{R}^n$ such that on query $\mathbf{x} \in \mathbb{R}^n$ the oracle outputs a vector $\mathbf{c}$ and either $\mathbf{c} = \mathbf{0}$, in which case $\mathbf{x} \in K$, or $\mathbf{c} \neq \mathbf{0}$, in which case $H \coloneqq \{\mathbf{z} \colon \mathbf{c}^\top \mathbf{z} \leq \mathbf{c}^\top \mathbf{x}\} \supset K$. The goal is to query a point $\mathbf{x} \in K$.*

Jiang et al. (2020) developed a cutting plane method that solves Problem 2 using $O(n \log(nR/r))$ queries to a separation oracle where $R$ and $r$ are parameters related to the convex set $\mathcal{K}$.

**Lemma 3** (Theorem 1.1, Jiang et al. (2020))**.** *There is a cutting plane method which solves Problem 2 using at most $C \cdot n \log(nR/r)$ queries for some constant $C$, given that the set $K$ is contained in the ball of radius $R$ centered at the origin and it contains a ball of radius $r$.*

Nemirovski (1994); Lee et al. (2015) showed that, running cutting plane method on a Lipschitz convex function $f$ with the separation oracle being the gradient of $f$ would yield a sequence of points where at least one of them is $\epsilon$-optimal. Furthermore, Sidford & Zhang (2023) showed that even if we cannot access the exact gradient value of $f$, it suffices to use an approximate gradient estimate with absolute error at most $O(\epsilon/R)$.

In this work, we show that this result can be extended to the case where we have an estimate of the gradient direction instead of the gradient itself. Specifically, we prove the following result.

**Theorem 3.** *There exists an algorithm based on cutting plane method that solves Problem 1 using $O(n^2 \log(nLR^2/\epsilon))$ queries.*

Note that Theorem 3 improves the prior state-of-the-art from $\tilde{O}(n^4)$ by Karabag et al. (2021) to $\tilde{O}(n^2)$.

*Proof of Theorem 3.* The proof follows a similar intuition as the proof of Proposition 1 in Sidford & Zhang (2023). Define $\mathcal{K}_{\epsilon/2}$ to be the set of $\epsilon/2$-optimal points of $f$, and $\mathcal{K}_{\epsilon}$ to be the set of $\epsilon$-optimal points of $f$. Given that $f$ is $L$-smooth, $\mathcal{K}_{\epsilon/2}$ must contain a ball of radius at least $r_{\mathcal{K}} = \sqrt{\epsilon/L}$ since for any $\mathbf{x}$ with $\|\mathbf{x} - \mathbf{x}^*\| \leq r_{\mathcal{K}}$ we have

$$f(\mathbf{x}) - f(\mathbf{x}^*) \leq L\|\mathbf{x} - \mathbf{x}^*\|^2/2 \leq \epsilon/2.$$

We apply the cutting plane method, as described in Lemma 3, to query a point in $\mathcal{K}_{\epsilon/2}$, which is a subset of the ball $\mathbb{B}_{2R}(\mathbf{0})$. To achieve this, at each query $\mathbf{x}$ of the cutting plane method, we use Comparison-GDE$(\mathbf{x}, \delta, \gamma)$, our comparison-based gradient direction estimation algorithm (Algorithm 2), as the separation oracle for the cutting plane method, where we set

$$\delta = \frac{1}{16R}\sqrt{\frac{\epsilon}{L}}, \qquad \gamma = \sqrt{2L\epsilon}.$$

We show that any query outside of $\mathcal{K}_{\epsilon}$ to Comparison-GDE$(\mathbf{x}, \delta, \gamma)$ will be a valid separation oracle for $\mathcal{K}_{\epsilon/2}$. In particular, if we ever queried Comparison-GDE$(\mathbf{x}, \delta, \gamma)$ at any $\mathbf{x} \in \mathbb{B}_{2R}(\mathbf{0}) \setminus \mathcal{K}_{\epsilon}$ with output being $\hat{\mathbf{g}}$, for any $\mathbf{y} \in \mathcal{K}_{\epsilon/2}$ we have

$$\langle \hat{\mathbf{g}}, \mathbf{y} - \mathbf{x} \rangle \leq \left\langle \frac{\nabla f(\mathbf{x})}{\|\nabla f(\mathbf{x})\|}, \mathbf{y} - \mathbf{x} \right\rangle + \left\| \hat{\mathbf{g}} - \frac{\nabla f(\mathbf{x})}{\|\nabla f(\mathbf{x})\|} \right\| \cdot \|\mathbf{y} - \mathbf{x}\|$$

$$\leq \frac{f(\mathbf{y}) - f(\mathbf{x})}{\|\nabla f(\mathbf{x})\|} + \left\| \hat{\mathbf{g}} - \frac{\nabla f(\mathbf{x})}{\|\nabla f(\mathbf{x})\|} \right\| \cdot \|\mathbf{y} - \mathbf{x}\| \leq -\frac{\epsilon}{2} + \frac{\epsilon}{10R} \cdot 4R < 0,$$

where

$$\|\nabla f(\mathbf{x})\| \geq (f(\mathbf{x}) - f^*)/\|\mathbf{x} - \mathbf{x}^*\| \geq (f(\mathbf{x}) - f^*)/(2R)$$

given that $f$ is convex. Combined with Theorem 1, it guarantees that

$$\left\| \hat{\mathbf{g}} - \frac{\nabla f(\mathbf{x})}{\|\nabla f(\mathbf{x})\|} \right\| \leq \delta = \frac{1}{16R}\sqrt{\frac{\epsilon}{L}}.$$

Hence,

$$\langle \hat{\mathbf{g}}, \mathbf{y} - \mathbf{x} \rangle \leq \frac{f(\mathbf{y}) - f(\mathbf{x})}{\|\nabla f(\mathbf{x})\|} + \left\| \hat{\mathbf{g}} - \frac{\nabla f(\mathbf{x})}{\|\nabla f(\mathbf{x})\|} \right\| \cdot \|\mathbf{y} - \mathbf{x}\| \leq -\frac{1}{2}\sqrt{\frac{\epsilon}{2L}} + \frac{1}{16R}\sqrt{\frac{\epsilon}{L}} \cdot 4R < 0,$$

indicating that $\hat{\mathbf{g}}$ is a valid separation oracle for the set $\mathcal{K}_{\epsilon/2}$. Consequently, by Lemma 3, after $Cn \log(nR/r_{\mathcal{K}})$ iterations, at least one of the queries must lie within $\mathcal{K}_{\epsilon}$, and we can choose the query with minimum function value to output, which can be done by making $Cn \log(nR/r_{\mathcal{K}})$ comparisons.

Note that in each iteration $O(n \log(n/\delta))$ queries to $O_f^{\text{Comp}}$ (1) are needed. Hence, the overall query complexity equals

$$Cn \log(nR/r_{\mathcal{K}}) \cdot O(n \log(n/\delta)) + Cn \log(nR/r_{\mathcal{K}}) = O\left(n^2 \log\left(nLR^2/\epsilon\right)\right).$$

$\square$

---

**Algorithm 4:** Comparison-based Approximate Normalized Gradient Descent (Comparison-NGD)

---

**Input:** Function $f \colon \mathbb{R}^n \to \mathbb{R}$, $\Delta$, precision $\epsilon$

**1** $T \leftarrow \frac{18L\Delta}{\epsilon^2}$, $\mathbf{x}_0 \leftarrow \mathbf{0}$

**2 for** $t = 0, \ldots, T-1$ **do**

**3**  $\quad \hat{\mathbf{g}}_t \leftarrow$ Comparison-GDE$(\mathbf{x}_t, 1/6, \epsilon/12)$

**4**  $\quad \mathbf{x}_t = \mathbf{x}_{t-1} - \epsilon \hat{\mathbf{g}}_t/(3L)$

**5** Uniformly randomly select $\mathbf{x}_{\mathrm{out}}$ from $\{\mathbf{x}_0, \ldots, \mathbf{x}_T\}$

**6 return** $\mathbf{x}_{\mathrm{out}}$

---

## 4   NONCONVEX OPTIMIZATION BY COMPARISONS

In this section, we study nonconvex optimization with function value comparisons. We first develop an algorithm that finds an $\epsilon$-FOSP of a smooth nonconvex function in Section 4.1. Then in Section 4.2, we further develop an algorithm that finds an $\epsilon$-SOSP of a nonconvex function that is smooth and Hessian-Lipschitz.

### 4.1   FIRST-ORDER STATIONARY POINT COMPUTATION BY COMPARISONS

In this subsection, we focus on the problem of finding an $\epsilon$-FOSP of a smooth nonconvex function by making function value comparisons.

**Problem 3** (Comparison-based first-order stationary point computation). *In the Comparison-based first-order stationary point computation (Comparison-FOSP) problem we are given query access to a comparison oracle $O_f^{\mathrm{Comp}}$ (1) for an L-smooth (possibly) nonconvex function $f \colon \mathbb{R}^n \to \mathbb{R}$ satisfying $f(\mathbf{0}) - \inf_{\mathbf{x}} f(\mathbf{x}) \leq \Delta$. The goal is to output an $\epsilon$-FOSP of $f$.*

We develop a comparison-based normalized gradient descent algorithm that solves Problem 3.

**Theorem 4.** *With success probability at least* $2/3$, *Algorithm 4 solves Problem 3 using* $O(L\Delta n \log n/\epsilon^2)$ *queries.*

The proof of Theorem 4 is deferred to Appendix C.1.

### 4.2   ESCAPING SADDLE POINTS OF NONCONVEX FUNCTIONS BY COMPARISONS

In this subsection, we focus on the problem of escaping from saddle points, i.e., finding an $\epsilon$-SOSP of a nonconvex function that is smooth and Hessian-Lipschitz, by making function value comparisons.

**Problem 4** (Comparison-based escaping from saddle point). *In the Comparison-based escaping from saddle point (Comparison-SOSP) problem we are given query access to a comparison oracle $O_f^{\mathrm{Comp}}$ (1) for a (possibly) nonconvex function $f \colon \mathbb{R}^n \to \mathbb{R}$ satisfying $f(\mathbf{0}) - \inf_{\mathbf{x}} f(\mathbf{x}) \leq \Delta$ that is L-smooth and $\rho$-Hessian Lipschitz. The goal is to output an $\epsilon$-SOSP of $f$.*

Our algorithm for Problem 4 given in Algorithm 5 is a combination of comparison-based normalized gradient descent and comparison-based negative curvature descent (Comparison-NCD). Specifically, Comparison-NCD is built upon our comparison-based negative curvature finding algorithms, Comparison-NCF1 (Algorithm 8) and Comparison-NCF2 (Algorithm 9) that work when the gradient is small or large respectively, and can decrease the function value efficiently when applied at a point with a large negative curvature.

**Lemma 4.** *In the setting of Problem 4, for any $\mathbf{z}$ satisfying $\lambda_{\min}(\nabla^2 f(\mathbf{x})) \leq -\sqrt{\rho\epsilon}$, Algorithm 6 outputs a point $\mathbf{z}_{\mathrm{out}} \in \mathbb{R}^n$ satisfying*

$$f(\mathbf{z}_{\mathrm{out}}) - f(\mathbf{z}) \leq -\frac{1}{48}\sqrt{\frac{\epsilon^3}{\rho}}$$

*with success probability at least $1 - \zeta$ using $O\big(\frac{L^2 n^{3/2}}{\zeta \rho \epsilon} \log^2 \frac{nL}{\zeta\sqrt{\rho\epsilon}}\big)$ queries.*

---

**Algorithm 5:** Comparison-based Perturbed Normalized Gradient Descent (Comparison-PNGD)

**Input:** Function $f\colon \mathbb{R}^n \to \mathbb{R}$, $\Delta$, precision $\epsilon$

1   $\mathcal{S} \leftarrow 350\Delta\sqrt{\frac{\rho}{\epsilon^3}}$, $\delta \leftarrow \frac{1}{6}$, $\mathbf{x}_{1,0} \leftarrow \mathbf{0}$

2   $\mathcal{T} \leftarrow \frac{384 L^2 \sqrt{n}}{\delta\rho\epsilon} \log \frac{36nL}{\sqrt{\rho\epsilon}}$, $p \leftarrow \frac{100}{\mathcal{T}} \log \mathcal{S}$

3   **for** $s = 1, \ldots, \mathcal{S}$ **do**

4     **for** $t = 0, \ldots, \mathcal{T} - 1$ **do**

5       $\hat{\mathbf{g}}_t \leftarrow$ Comparison-GDE$(\mathbf{x}_{s,t}, \delta, \gamma)$

6       $\mathbf{y}_{s,t} \leftarrow \mathbf{x}_{s,t} - \epsilon\hat{\mathbf{g}}_t/(3L)$

7       Choose $\mathbf{x}_{s,t+1}$ to be the point between $\{x_{s,t}, \mathbf{y}_{s,t}\}$ with smaller function value

8       $\mathbf{x}'_{s,t+1} \leftarrow \begin{cases} \mathbf{0}, \text{ w.p. } 1-p \\ \text{Comparison-NCD}(\mathbf{x}_{s,t+1}, \epsilon, \delta), \text{ w.p. } p \end{cases}$

9     Choose $\mathbf{x}_{s+1,0}$ among $\{\mathbf{x}_{s,0}, \ldots, \mathbf{x}_{s,\mathcal{T}}, \mathbf{x}'_{s,0}, \ldots, \mathbf{x}'_{s,\mathcal{T}}\}$ with the smallest function value.

10     $\mathbf{x}'_{s+1,0} \leftarrow \begin{cases} \mathbf{0}, \text{ w.p. } 1-p \\ \text{Comparison-NCD}(\mathbf{x}_{s+1,0}, \epsilon, \delta), \text{ w.p. } p \end{cases}$

11   Uniformly randomly select $s_{\text{out}} \in \{1, \ldots, \mathcal{S}\}$ and $t_{\text{out}} \in [\mathcal{T}]$

12   **return** $\mathbf{x}_{s_{\text{out}}, t_{\text{out}}}$

---

**Algorithm 6:** Comparison-based Negative Curvature Descent (Comparison-NCD)

**Input:** Function $f\colon \mathbb{R}^n \to \mathbb{R}$, precision $\epsilon$, input point $\mathbf{z}$, error probability $\delta$

1   $\mathbf{v}_1 \leftarrow$ Comparison-NCF1$(\mathbf{z}, \epsilon, \delta)$

2   $\mathbf{v}_2 \leftarrow$ Comparison-NCF2$(\mathbf{z}, \epsilon, \delta)$

3   $\mathbf{z}_{1,+} = \mathbf{z} + \frac{1}{2}\sqrt{\frac{\epsilon}{\rho}}\mathbf{v}_1$, $\mathbf{z}_{1,-} = \mathbf{z} - \frac{1}{2}\sqrt{\frac{\epsilon}{\rho}}\mathbf{v}_1$, $\mathbf{z}_{2,+} = \mathbf{z} + \frac{1}{2}\sqrt{\frac{\epsilon}{\rho}}\mathbf{v}_2$, $\mathbf{z}_{2,-} = \mathbf{z} - \frac{1}{2}\sqrt{\frac{\epsilon}{\rho}}\mathbf{v}_2$

4   **return** $\mathbf{z}_{\text{out}} \in \{\mathbf{z}_{1,+}, \mathbf{z}_{1,-}, \mathbf{z}_{2,+}, \mathbf{z}_{2,-}\}$ with the smallest function value.

---

The proof of Lemma 4 is deferred to Appendix C.3. Next, we present the main result of this subsection, which describes the complexity of solving Problem 4 using Algorithm 5.

**Theorem 5.** *With success probability at least* $2/3$, *Algorithm 5 solves Problem 4 using an expected* $O\big(\frac{\Delta L^2 n^{3/2}}{\rho^{1/2}\epsilon^{5/2}} \log^3 \frac{nL}{\sqrt{\rho\epsilon}}\big)$ *queries.*

The proof of Theorem 5 is deferred to Appendix C.4.

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
