## A AUXILIARY LEMMAS

### A.1 DISTANCE BETWEEN NORMALIZED VECTORS

**Lemma 5.** *If* $\mathbf{v}, \mathbf{v}' \in \mathbb{R}^n$ *are two vectors such that* $\|\mathbf{v}\| \geq \gamma$ *and* $\|\mathbf{v} - \mathbf{v}'\| \leq \tau$, *we have*

$$\left\| \frac{\mathbf{v}}{\|\mathbf{v}\|} - \frac{\mathbf{v}'}{\|\mathbf{v}'\|} \right\| \leq \frac{2\tau}{\gamma}.$$

*Proof.* By the triangle inequality, we have

$$
\left\| \frac{\mathbf{v}}{\|\mathbf{v}\|} - \frac{\mathbf{v}'}{\|\mathbf{v}'\|} \right\| \leq \left\| \frac{\mathbf{v}}{\|\mathbf{v}\|} - \frac{\mathbf{v}'}{\|\mathbf{v}\|} \right\| + \left\| \frac{\mathbf{v}'}{\|\mathbf{v}\|} - \frac{\mathbf{v}'}{\|\mathbf{v}'\|} \right\|
$$

$$
= \frac{\|\mathbf{v} - \mathbf{v}'\|}{\|\mathbf{v}\|} + \frac{|\|\mathbf{v}\| - \|\mathbf{v}'\|| \|\mathbf{v}'\|}{\|\mathbf{v}\| \|\mathbf{v}'\|}
$$

$$
\leq \frac{\tau}{\gamma} + \frac{\tau}{\gamma} = \frac{2\tau}{\gamma}.
$$

$\square$

**Lemma 6.** *If* $\mathbf{v}_1, \mathbf{v}_2 \in \mathbb{R}^n$ *are two vectors such that* $\|\mathbf{v}_1\|, \|\mathbf{v}_2\| \geq \gamma$, *and* $\mathbf{v}'_1, \mathbf{v}'_2 \in \mathbb{R}^n$ *are another two vectors such that* $\|\mathbf{v}_1 - \mathbf{v}'_1\|, \|\mathbf{v}_2 - \mathbf{v}'_2\| \leq \tau$ *where* $0 < \tau < \gamma$, *we have*

$$\left| \left\langle \frac{\mathbf{v}_1}{\|\mathbf{v}_1\|}, \frac{\mathbf{v}_2}{\|\mathbf{v}_2\|} \right\rangle - \left\langle \frac{\mathbf{v}'_1}{\|\mathbf{v}'_1\|}, \frac{\mathbf{v}'_2}{\|\mathbf{v}'_2\|} \right\rangle \right| \leq \frac{6\tau}{\gamma}.$$

*Proof.* By the triangle inequality, we have

$$
\left| \left\langle \frac{\mathbf{v}_1}{\|\mathbf{v}_1\|}, \frac{\mathbf{v}_2}{\|\mathbf{v}_2\|} \right\rangle - \left\langle \frac{\mathbf{v}'_1}{\|\mathbf{v}'_1\|}, \frac{\mathbf{v}'_2}{\|\mathbf{v}'_2\|} \right\rangle \right|
$$

$$
\leq \left| \left\langle \frac{\mathbf{v}_1}{\|\mathbf{v}_1\|}, \frac{\mathbf{v}_2}{\|\mathbf{v}_2\|} \right\rangle - \left\langle \frac{\mathbf{v}'_1}{\|\mathbf{v}_1\|}, \frac{\mathbf{v}'_2}{\|\mathbf{v}_2\|} \right\rangle \right| + \left| \left\langle \frac{\mathbf{v}'_1}{\|\mathbf{v}_1\|}, \frac{\mathbf{v}'_2}{\|\mathbf{v}_2\|} \right\rangle - \left\langle \frac{\mathbf{v}'_1}{\|\mathbf{v}'_1\|}, \frac{\mathbf{v}'_2}{\|\mathbf{v}'_2\|} \right\rangle \right|.
$$

On the one hand, by the triangle inequality and the Cauchy-Schwarz inequality,

$$
\left| \left\langle \frac{\mathbf{v}_1}{\|\mathbf{v}_1\|}, \frac{\mathbf{v}_2}{\|\mathbf{v}_2\|} \right\rangle - \left\langle \frac{\mathbf{v}'_1}{\|\mathbf{v}_1\|}, \frac{\mathbf{v}'_2}{\|\mathbf{v}_2\|} \right\rangle \right| \leq \frac{1}{\|\mathbf{v}_1\| \|\mathbf{v}_2\|} (|\langle \mathbf{v}_1, \mathbf{v}_2 \rangle - \langle \mathbf{v}_1, \mathbf{v}'_2 \rangle| + |\langle \mathbf{v}_1, \mathbf{v}'_2 \rangle - \langle \mathbf{v}'_1, \mathbf{v}'_2 \rangle|)
$$

$$
\leq \frac{\|\mathbf{v}_2 - \mathbf{v}'_2\|}{\|\mathbf{v}_2\|} + \frac{\|\mathbf{v}_1 - \mathbf{v}'_1\| \|\mathbf{v}'_2\|}{\|\mathbf{v}_1\| \|\mathbf{v}_2\|}
$$

$$
\leq \frac{\tau}{\gamma} + \frac{\tau(\gamma + \tau)}{\gamma^2}.
$$

On the other hand, by the Cauchy-Schwarz inequality, $|\langle \mathbf{v}'_1, \mathbf{v}'_2 \rangle| \leq \|\mathbf{v}'_1\| \|\mathbf{v}'_2\|$, and hence

$$
\left| \left\langle \frac{\mathbf{v}'_1}{\|\mathbf{v}_1\|}, \frac{\mathbf{v}'_2}{\|\mathbf{v}_2\|} \right\rangle - \left\langle \frac{\mathbf{v}'_1}{\|\mathbf{v}'_1\|}, \frac{\mathbf{v}'_2}{\|\mathbf{v}'_2\|} \right\rangle \right| = |\langle \mathbf{v}'_1, \mathbf{v}'_2 \rangle| \left| \frac{1}{\|\mathbf{v}_1\| \|\mathbf{v}_2\|} - \frac{1}{\|\mathbf{v}'_1\| \|\mathbf{v}'_2\|} \right|
$$

$$
\leq \left| \frac{\|\mathbf{v}'_1\| \|\mathbf{v}'_2\|}{\|\mathbf{v}_1\| \|\mathbf{v}_2\|} - 1 \right|
$$

$$
\leq \left( \frac{\gamma + \tau}{\gamma} \right)^2 - 1.
$$

In all, due to $\tau < \gamma$,

$$
\left| \left\langle \frac{\mathbf{v}_1}{\|\mathbf{v}_1\|}, \frac{\mathbf{v}_2}{\|\mathbf{v}_2\|} \right\rangle - \left\langle \frac{\mathbf{v}'_1}{\|\mathbf{v}'_1\|}, \frac{\mathbf{v}'_2}{\|\mathbf{v}'_2\|} \right\rangle \right| \leq \frac{\tau}{\gamma} + \frac{\tau(\gamma + \tau)}{\gamma^2} + \left( \frac{\gamma + \tau}{\gamma} \right)^2 - 1 = \frac{2\tau(2\gamma + \tau)}{\gamma^2} \leq \frac{6\tau}{\gamma}.
$$

$\square$

## A.2 A FACT FOR VECTOR NORMS

**Lemma 7.** *For any nonzero vectors* $\mathbf{v}, \mathbf{g} \in \mathbb{R}^n$,

$$\sqrt{\frac{1 - \langle \frac{\mathbf{v}+\mathbf{g}}{\|\mathbf{v}+\mathbf{g}\|}, \frac{\mathbf{v}}{\|\mathbf{v}\|} \rangle^2}{1 - \langle \frac{\mathbf{v}-\mathbf{g}}{\|\mathbf{v}-\mathbf{g}\|}, \frac{\mathbf{v}}{\|\mathbf{v}\|} \rangle^2}} = \frac{\|\mathbf{v} - \mathbf{g}\|}{\|\mathbf{v} + \mathbf{g}\|}.$$

*Proof.* We have

$$\frac{1 - \langle \frac{\mathbf{v}+\mathbf{g}}{\|\mathbf{v}+\mathbf{g}\|}, \frac{\mathbf{v}}{\|\mathbf{v}\|} \rangle^2}{1 - \langle \frac{\mathbf{v}-\mathbf{g}}{\|\mathbf{v}-\mathbf{g}\|}, \frac{\mathbf{v}}{\|\mathbf{v}\|} \rangle^2} \cdot \frac{\|\mathbf{v} + \mathbf{g}\|^2}{\|\mathbf{v} - \mathbf{g}\|^2} = \frac{\|\mathbf{v} + \mathbf{g}\|^2 - \langle \mathbf{v} + \mathbf{g}, \frac{\mathbf{v}}{\|\mathbf{v}\|} \rangle^2}{\|\mathbf{v} - \mathbf{g}\|^2 - \langle \mathbf{v} - \mathbf{g}, \frac{\mathbf{v}}{\|\mathbf{v}\|} \rangle^2}$$

$$= \frac{\langle \mathbf{v} + \mathbf{g}, \mathbf{v} + \mathbf{g} \rangle - (\|\mathbf{v}\| + \frac{\langle \mathbf{v}, \mathbf{g} \rangle}{\|\mathbf{v}\|})^2}{\langle \mathbf{v} - \mathbf{g}, \mathbf{v} - \mathbf{g} \rangle - (\|\mathbf{v}\| - \frac{\langle \mathbf{v}, \mathbf{g} \rangle}{\|\mathbf{v}\|})^2}$$

$$= \frac{\|\mathbf{v}\|^2 + \|\mathbf{g}\|^2 + 2\langle \mathbf{v}, \mathbf{g} \rangle - (\|\mathbf{v}\|^2 + 2\langle \mathbf{v}, \mathbf{g} \rangle + \frac{\langle \mathbf{v}, \mathbf{g} \rangle^2}{\|\mathbf{v}\|^2})}{\|\mathbf{v}\|^2 + \|\mathbf{g}\|^2 - 2\langle \mathbf{v}, \mathbf{g} \rangle - (\|\mathbf{v}\|^2 - 2\langle \mathbf{v}, \mathbf{g} \rangle + \frac{\langle \mathbf{v}, \mathbf{g} \rangle^2}{\|\mathbf{v}\|^2})}$$

$$= 1.$$

$\square$

## A.3 GRADIENT UPPER BOUND OF SMOOTH CONVEX FUNCTIONS

**Lemma 8** (Lemma A.2, Levy (2017)). *For any $L$-smooth convex function $f : \mathbb{R}^n \to \mathbb{R}$ and any* $\mathbf{x} \in \mathbb{R}^n$, *we have*

$$\|\nabla f(\mathbf{x})\|^2 \leq 2L(f(\mathbf{x}) - f^*).$$

# B APPROXIMATE ADAPTIVE NORMALIZED GRADIENT DESCENT (APPROX-ADANGD)

In this section, we prove technical details of the normalized gradient descent we use for convex optimization. Inspired by Levy (2017) which condcuted a detailed analysis for the normalized gradient descent method, we first introduce the Approximate Adaptive Gradient Descent (Approx-AdaGrad) algorithm below:

---

**Algorithm 7:** Approximate Adaptive Gradient Descent (Approx-AdaGrad)

---

**Input:** # Iterations $T$, a set of convex functions $\{f_t\}_{t=1}^T$, $\mathbf{x}_0 \in \mathbb{R}^n$, a convex set $\mathcal{K}$ with diameter $D$

1 **for** $t = 1, \ldots, T$ **do**
2     Calculate an estimate $\tilde{\mathbf{g}}_t$ of $\nabla f_t(\mathbf{x}_{t-1})$
3     $\eta_t \leftarrow D/\sqrt{2t}$
4     $\mathbf{x}_t = \Pi_{\mathcal{K}}(\mathbf{x}_{t-1} - \eta_t \tilde{\mathbf{g}}_t)$

---

**Lemma 9.** *Algorithm 7 guarantees the following regret*

$$\sum_{t=1}^T f_t(\mathbf{x}_t) - \min_{\mathbf{x} \in \mathcal{K}} \sum_{t=1}^T f_t(\mathbf{x}) \leq D\sqrt{2T} + T\delta D.$$

*if at each step $t$ we have*

$$\|\nabla f_t(\mathbf{x}_t)\| = 1, \quad \|\tilde{\mathbf{g}}_t - \nabla f_t(\mathbf{x}_t)\| \leq \delta, \quad \|\tilde{\mathbf{g}}_t\| = 1.$$

*Proof.* The proof follows the flow of the proof of Theorem 1.1 in Levy (2017). For any $t \in [T]$ and $\mathbf{x} \in \mathcal{K}$ we have

$$\|\mathbf{x}_{t+1} - \mathbf{x}\|^2 \leq \|\mathbf{x}_t - \mathbf{x}\|^2 - 2\eta_t \langle \tilde{\mathbf{g}}_t, \mathbf{x}_t - \mathbf{x} \rangle + \eta_t^2 \|\tilde{\mathbf{g}}_t\|^2$$

and

$$\langle \tilde{\mathbf{g}}_t, \mathbf{x}_t - \mathbf{x} \rangle \leq \frac{1}{2\eta_t} \left( \|\mathbf{x}_t - \mathbf{x}\|^2 - \|\mathbf{x}_{t+1} - \mathbf{x}\|^2 \right) + \frac{\eta_t}{2} \|\tilde{\mathbf{g}}_t\|^2.$$

Since $f_t$ is convex for each $t$, we have

$$\begin{aligned} f_t(\mathbf{x}_t) - f_t(\mathbf{x}) &\leq \langle \nabla f_t(\mathbf{x}_t), \mathbf{x}_t - \mathbf{x} \rangle \\ &\leq \langle \tilde{\mathbf{g}}_t, \mathbf{x}_t - \mathbf{x} \rangle + \|\tilde{\mathbf{g}}_t - \nabla f_t(\mathbf{x}_t)\| \cdot \|\mathbf{x}_t - \mathbf{x}\| \\ &\leq \langle \tilde{\mathbf{g}}_t, \mathbf{x}_t - \mathbf{x} \rangle + \delta D, \end{aligned}$$

which leads to

$$\sum_{t=1}^T f_t(\mathbf{x}_t) - \sum_{t=1}^T f_t(\mathbf{x}) \leq \sum_{t=1}^T \frac{\|\mathbf{x}_t - \mathbf{x}\|^2}{2} \left( \frac{1}{\eta_t} - \frac{1}{\eta_{t-1}} \right) + \sum_{t=1}^T \frac{\eta_t}{2} \|\tilde{\mathbf{g}}_t\|^2 + T\delta D,$$

where we denote $\eta_0 = \infty$. Further we can derive that

$$\begin{aligned} \sum_{t=1}^T f_t(\mathbf{x}_t) - \sum_{t=1}^T f_t(\mathbf{x}) &\leq \frac{D^2}{2} \sum_{t=1}^T \left( \frac{1}{\eta_t} - \frac{1}{\eta_{t-1}} \right) + \frac{D}{2\sqrt{2}} \sum_{t=1}^T \frac{\|\tilde{\mathbf{g}}_t\|^2}{\sqrt{t}} + T\delta D \\ &\leq \frac{D^2}{2\eta_T} + \frac{D}{2\sqrt{2}} \sum_{t=1}^T \frac{1}{\sqrt{t}} + T\delta D, \end{aligned}$$

Moreover, we have

$$\sum_{t=1}^T \frac{1}{\sqrt{t}} \leq 2\sqrt{T},$$

which leads to

$$\begin{aligned} \sum_{t=1}^T f_t(\mathbf{x}_t) - \sum_{t=1}^T f_t(\mathbf{x}) &\leq \frac{D^2}{2\eta_T} + \frac{D}{2\sqrt{2}} \sum_{t=1}^T \frac{1}{\sqrt{t}} + T\delta D \\ &\leq D\sqrt{2T} + T\delta D. \end{aligned}$$

$\square$

Now, we can prove Lemma 2 which guarantees the completeness of Theorem 2.

*Proof of Lemma 2.* The proof follows the flow of the proof of Theorem 2.1 in Levy (2017). In particular, observe that Algorithm 3 is equivalent to applying Approx-AdaGrad (Algorithm 7) to the following sequence of functions

$$\tilde{f}_t(\mathbf{x}) := \frac{\langle \nabla f(\mathbf{x}_t), \mathbf{x} \rangle}{\|\nabla f(\mathbf{x}_t)\|}, \quad \forall t \in [T].$$

Then by Lemma 9, for any $\mathbf{x} \in \mathcal{K}$ we have

$$\sum_{t=1}^T \frac{\langle \nabla f(\mathbf{x}_t), \mathbf{x}_t - \mathbf{x} \rangle}{\|\nabla f(\mathbf{x}_t)\|} \leq D\sqrt{2T} + T\delta D,$$

where

$$f(\mathbf{x}_t) - f(\mathbf{x}) \leq \langle \nabla f(\mathbf{x}_t), \mathbf{x}_t - \mathbf{x} \rangle, \quad \forall t \in [T]$$

given that $f$ is convex, and $D = 2R$ is the diameter of $\mathbb{B}_R(\mathbf{0})$. Hence,

$$\min_{t \in [T]} f(\mathbf{x}_t) - f^* \leq \frac{\sum_{t=1}^T (f(\mathbf{x}_t) - f^*)/\|\nabla f(\mathbf{x}_t)\|}{\sum_{t=1}^T 1/\|\nabla f(\mathbf{x}_t)\|} \leq \frac{2R\sqrt{2T} + 2T\delta R}{\sum_{t=1}^T 1/\|\nabla f(\mathbf{x}_t)\|}.$$

Next, we proceed to bound the term $\sum_{t=1}^{T} 1/\|\nabla f(\mathbf{x}_t)\|$ on the denominator. By the Cauchy-Schwarz inequality,

$$\left( \sum_{t=1}^{T} 1/\|\nabla f(\mathbf{x}_t)\| \right) \cdot \left( \sum_{t=1}^{T} \|\nabla f(\mathbf{x}_t)\| \right) \geq T^2,$$

which leads to

$$\sum_{t=1}^{T} \frac{1}{\|\nabla f(\mathbf{x}_t)\|} \geq \frac{T^2}{\sum_{t=1}^{T} \|\nabla f(\mathbf{x}_t)\|},$$

where

$$\begin{aligned}
\sum_{t=1}^{T} \|\nabla f(\mathbf{x}_t)\| &= \sum_{t=1}^{T} \frac{\|\nabla f(\mathbf{x}_t)\|^2}{\|\nabla f(\mathbf{x}_t)\|} \\
&\leq \sum_{t=1}^{T} \frac{2L(f(\mathbf{x}_t) - f^*)}{\|\nabla f(\mathbf{x}_t)\|} \\
&\leq 2L \sum_{t=1}^{T} \frac{\langle \nabla f(\mathbf{x}_t), \mathbf{x}_t - \mathbf{x}^* \rangle}{\|\nabla f(\mathbf{x}_t)\|} \\
&\leq 2L(2R\sqrt{2T} + 2T\delta R),
\end{aligned}$$

where the first inequality is by Lemma 8, the second inequality is by the convexity of $f$, and the third inequality is due to Lemma 9. Further we can derive that

$$\min_{t \in [T]} f(\mathbf{x}_t) - f^* \leq \frac{8R\sqrt{T} + 2T\delta R}{\sum_{t=1}^{T} 1/\|\nabla f(\mathbf{x}_t)\|} \leq \frac{2L(2R\sqrt{2T} + 2T\delta R)^2}{T^2}.$$

$\square$

## C  PROOF DETAILS OF NONCONVEX OPTIMIZATION BY COMPARISONS

### C.1  PROOF OF THEOREM 4

*Proof of Theorem 4.* We prove the correctness of Theorem 4 by contradiction. For any iteration $t \in [T]$ with $\|\nabla f(\mathbf{x}_t)\| > \epsilon$, by Theorem 1 we have

$$\left\| \hat{\mathbf{g}}_t - \frac{\nabla f(\mathbf{x}_t)}{\|\nabla f(\mathbf{x}_t)\|} \right\| \leq \delta = \frac{1}{6},$$

indicating

$$\begin{aligned}
f(\mathbf{x}_{t+1}) - f(\mathbf{x}_t) &\leq \langle \nabla f(\mathbf{x}_t), \mathbf{x}_{t+1} - \mathbf{x}_t \rangle + \frac{L}{2}\|\mathbf{x}_{t+1} - \mathbf{x}_t\|^2 \\
&\leq -\frac{\epsilon}{3L}\langle \nabla f(\mathbf{x}_t), \hat{\mathbf{g}}_t \rangle + \frac{L}{2}\left( \frac{\epsilon}{3L} \right)^2 \\
&\leq -\frac{\epsilon}{3L}\|\nabla f(\mathbf{x}_t)\|(1-\delta) + \frac{\epsilon^2}{18L} \leq -\frac{2\epsilon^2}{9L}.
\end{aligned}$$

That is to say, for any iteration $t$ such that $\mathbf{x}_t$ is not an $\epsilon$-FOSP, the function value will decrease at least $\frac{2\epsilon^2}{9L}$ in this iteration. Furthermore, for any iteration $t \in [T]$ with $\frac{\epsilon}{12} < \|\nabla f(\mathbf{x}_t)\| \leq \epsilon$, by Theorem 1 we have

$$\left\| \hat{\mathbf{g}}_t - \frac{\nabla f(\mathbf{x}_t)}{\|\nabla f(\mathbf{x}_t)\|} \right\| \leq \delta = \frac{1}{6},$$

indicating

$$\begin{aligned}
f(\mathbf{x}_{t+1}) - f(\mathbf{x}_t) &\leq \langle \nabla f(\mathbf{x}_t), \mathbf{x}_{t+1} - \mathbf{x}_t \rangle + \frac{L}{2}\|\mathbf{x}_{t+1} - \mathbf{x}_t\|^2 \\
&\leq -\frac{\epsilon}{3L}\|\nabla f(\mathbf{x}_t)\|(1-\delta) + \frac{\epsilon^2}{18L} \leq 0. \quad (5)
\end{aligned}$$

For any iteration $t \in [T]$ with $\|\nabla f(\mathbf{x}_t)\| \leq \epsilon/12$, we have

$$f(\mathbf{x}_{t+1}) - f(\mathbf{x}_t) \leq \langle \nabla f(\mathbf{x}_t), \mathbf{x}_{t+1} - \mathbf{x}_t \rangle + \frac{L}{2}\|\mathbf{x}_{t+1} - \mathbf{x}_t\|^2$$

$$\leq \|\nabla f(\mathbf{x}_t)\| \cdot \|\mathbf{x}_{t+1} - \mathbf{x}_t\| + \frac{L}{2}\|\mathbf{x}_{t+1} - \mathbf{x}_t\|^2 \leq \frac{\epsilon^2}{12L}.$$

Combining (5) and the above inequality, we know that for any iteration $t$ such that $\mathbf{x}_t$ is an $\epsilon$-FOSP, the function value increases at most $\epsilon^2/(12L)$ in this iteration. Moreover, since

$$f(\mathbf{0}) - f(\mathbf{x}_T) \leq f(\mathbf{0}) - f^* \leq \Delta,$$

we can conclude that at least $2/3$ of the iterations have $\mathbf{x}_t$ being an $\epsilon$-FOSP, and randomly outputting one of them solves Problem 3 with success probability at least $2/3$.

The query complexity of Algorithm 4 only comes from the gradient direction estimation step in Line 3, which equals

$$T \cdot O(n \log(n/\delta)) = O\left(L\Delta n \log n/\epsilon^2\right).$$

$\square$

## C.2 NEGATIVE CURVATURE FINDING BY COMPARISONS

In this subsection, we show how to find a negative curvature direction of a point $\mathbf{x}$ satisfying $\lambda_{\min}(\nabla^2 f(\mathbf{x})) \leq -\sqrt{\rho\epsilon}$ Observe that the Hessian matrix $\nabla^2 f(\mathbf{x})$ admits the following eigen-decomposition:

$$\nabla^2 f(\mathbf{x}) = \sum_{i=1}^{n} \lambda_i \mathbf{u}_i \mathbf{u}_i^\top, \tag{6}$$

where the vectors $\{\mathbf{u}_i\}_{i=1}^n$ forms an orthonormal basis of $\mathbb{R}^n$. Without loss of generality we assume the eigenvalues $\lambda_1, \lambda_2, \ldots, \lambda_n$ corresponding to $\mathbf{u}_1, \mathbf{u}_2, \ldots, \mathbf{u}_n$ satisfy

$$\lambda_1 \leq \lambda_2 \leq \cdots \leq \lambda_n, \tag{7}$$

where $\lambda_1 \leq -\sqrt{\rho\epsilon}$. Throughout this subsection, for any vector $\mathbf{v} \in \mathbb{R}^n$, we denote

$$\mathbf{v}_\perp := \mathbf{v} - \langle \mathbf{v}, \mathbf{u}_1 \rangle \mathbf{u}_1$$

to be the component of $\mathbf{v}$ that is orthogonal to $\mathbf{u}_1$.

### C.2.1 NEGATIVE CURVATURE FINDING WHEN THE GRADIENT IS RELATIVELY SMALL

In this part, we present our negative curvature finding algorithm that finds the negative curvature of a point $\mathbf{x}$ with $\lambda_{\min}(\nabla^2 f(\mathbf{x})) \leq -\sqrt{\rho\epsilon}$ when the norm of the gradient $\nabla f(\mathbf{x})$ is relatively small.

---

**Algorithm 8:** Comparison-based Negative Curvature Finding 1 (Comparison-NCF1)

**Input:** Function $f \colon \mathbb{R}^n \to \mathbb{R}$, $\mathbf{x}$, precision $\epsilon$, error probability $\delta$

1   $\mathscr{T} \leftarrow \frac{384L^2\sqrt{n}}{\delta\rho\epsilon}\log\frac{36nL}{\sqrt{\rho\epsilon}}, \hat{\delta} \leftarrow \frac{1}{8\mathscr{T}(\rho\epsilon)^{1/4}}\sqrt{\frac{\pi L}{n}}, r \leftarrow \frac{\pi\delta(\rho\epsilon)^{1/4}\sqrt{L}}{128\rho n\mathscr{T}}, \gamma \leftarrow \frac{\delta r}{16}\sqrt{\frac{\pi\rho\epsilon}{n}}$

2   $\mathbf{y}_0 \leftarrow \text{Uniform}(\mathcal{S}^{n-1})$

3   **for** $t = 0, \ldots, \mathscr{T} - 1$ **do**

4      $\hat{\mathbf{g}}_t \leftarrow \text{Comparison-GDE}(\mathbf{x} + r\mathbf{y}_t, \hat{\delta}, \gamma)$

5      $\bar{\mathbf{y}}_{t+1} \leftarrow \mathbf{y}_t - \frac{\delta}{16L}\sqrt{\frac{\rho\epsilon}{n}}\hat{\mathbf{g}}_t$

6      $\mathbf{y}_{t+1} \leftarrow \bar{\mathbf{y}}_{t+1}/\|\bar{\mathbf{y}}_{t+1}\|$

7   **return** $\hat{\mathbf{e}} \leftarrow \mathbf{y}_\mathscr{T}$

---

**Lemma 10.** *In the setting of Problem 4, for any* $\mathbf{x}$ *satisfying*

$$\|\nabla f(\mathbf{x})\| \le L \left( \frac{\pi\delta}{256n\mathscr{T}} \right)^2 \sqrt{\frac{\epsilon}{\rho}}, \qquad \lambda_{\min}(\nabla^2 f(\mathbf{x})) \le -\sqrt{\rho\epsilon},$$

*Algorithm 8 outputs a unit vector* $\hat{\mathbf{e}}$ *satisfying*

$$\hat{\mathbf{e}}^{\mathscr{T}} \nabla^2 f(\mathbf{x}) \hat{\mathbf{e}} \le -\sqrt{\rho\epsilon}/4,$$

*with success probability at least* $1 - \delta$ *using*

$$O\left( \frac{L^2 n^{3/2}}{\delta\rho\epsilon} \log^2 \frac{nL}{\delta\sqrt{\rho\epsilon}} \right)$$

*queries.*

To prove Lemma 10, without loss of generality we assume $\mathbf{x} = \mathbf{0}$ by shifting $\mathbb{R}^n$ such that $\mathbf{x}$ is mapped to $\mathbf{0}$. We denote $\mathbf{z}_t \coloneqq r\mathbf{y}_t/\|\mathbf{y}_t\|$ for each iteration $t \in [\mathscr{T}]$ of Algorithm 8.

**Lemma 11.** *In the setting of Problem 4, for any iteration* $t \in [\mathscr{T}]$ *of Algorithm 8 with* $|y_{t,1}| \ge \frac{\delta}{8}\sqrt{\frac{\pi}{n}}$, *we have*

$$\|\nabla f(\mathbf{z}_t)\| \ge \frac{\delta r}{16} \sqrt{\frac{\pi\rho\epsilon}{n}}.$$

*Proof.* Observe that

$$
\begin{aligned}
\|\nabla f(\mathbf{z}_k)\| &\ge |\nabla_1 f(\mathbf{z}_k)| \\
&= |\nabla_1 f(\mathbf{0}) + (\nabla^2 f(\mathbf{0})\mathbf{z}_k)_1 + \nabla_1 f(\mathbf{z}_k) - \nabla_1 f(\mathbf{0}) - (\nabla^2 f(\mathbf{0})\mathbf{z}_k)_1| \\
&\ge |(\nabla^2 f(\mathbf{0})\mathbf{z}_k)_1| - |\nabla_1 f(\mathbf{0})| - |\nabla_1 f(\mathbf{z}_k) - \nabla_1 f(\mathbf{0}) - (\nabla^2 f(\mathbf{0})\mathbf{z}_k)_1|.
\end{aligned}
$$

Given that $f$ is $\rho$-Hessian Lipschitz, we have

$$|\nabla_1 f(\mathbf{z}_k) - \nabla_1 f(\mathbf{0}) - (\nabla^2 f(\mathbf{0})\mathbf{z}_k)_1| \le \frac{\rho\|\mathbf{z}_k\|^2}{2} = \frac{\rho r^2}{2} \le \frac{\delta r}{32} \sqrt{\frac{\pi\rho\epsilon}{n}}.$$

Moreover, we have

$$|(\nabla^2 f(\mathbf{0})\mathbf{z}_k)_1| = \sqrt{\rho\epsilon}\|\mathbf{z}_{k,1}\| \ge \frac{\delta r}{8} \sqrt{\frac{\pi\rho\epsilon}{n}},$$

which leads to

$$
\begin{aligned}
\|\nabla f(\mathbf{z}_k)\| &\ge |\nabla_1 f(\mathbf{z}_k)| \\
&\ge |(\nabla^2 f(\mathbf{0})\mathbf{z}_k)_1| - |\nabla_1 f(\mathbf{0})| - |\nabla_1 f(\mathbf{z}_k) - \nabla_1 f(\mathbf{0}) - (\nabla^2 f(\mathbf{0})\mathbf{z}_k)_1| \\
&\ge \frac{\delta r}{16} \sqrt{\frac{\pi\rho\epsilon}{n}},
\end{aligned}
$$

where the last inequality is due to the fact that

$$\|\nabla_1 f(\mathbf{0})\| \le \|\nabla f(\mathbf{0})\| \le \frac{\pi\delta r(\rho\epsilon)^{1/4}\sqrt{L}}{256nT} \le \frac{\delta r}{32} \sqrt{\frac{\pi\rho\epsilon}{n}}.$$

$\square$

**Lemma 12.** *In the setting of Problem 4, for any iteration* $t \in [\mathscr{T}]$ *of Algorithm 8 we have*

$$|y_{t,1}| \ge \frac{\delta}{8} \sqrt{\frac{\pi}{n}} \tag{8}$$

*if* $|y_{0,1}| \ge \frac{\delta}{2}\sqrt{\frac{\pi}{n}}$ *and* $\|\nabla f(\mathbf{0})\| \le \frac{\delta r}{32}\sqrt{\frac{\pi\rho\epsilon}{n}}$.

*Proof.* We use recurrence to prove this lemma. In particular, assume

$$\frac{|y_{t,1}|}{\|\mathbf{y}_{t,\perp}\|} \geq \frac{\delta}{2}\sqrt{\frac{\pi}{n}}\left(1 - \frac{1}{2\mathscr{T}}\right)^t \tag{9}$$

is true for all $t \leq k$ for some $k$, which guarantees that

$$|y_{t,1}| \geq \frac{\delta}{4}\sqrt{\frac{\pi}{n}}\left(1 - \frac{1}{2\mathscr{T}}\right)^t$$

Then for $t = k + 1$, we have

$$\bar{\mathbf{y}}_{k+1,\perp} = \mathbf{y}_{k,\perp} - \frac{\delta}{16L}\sqrt{\frac{\rho\epsilon}{n}}\cdot\hat{\mathbf{g}}_{k,\perp},$$

and

$$\|\bar{\mathbf{y}}_{k+1,\perp}\| \leq \left\|\mathbf{y}_{k,\perp} - \frac{\delta}{16L}\sqrt{\frac{\rho\epsilon}{n}}\cdot\frac{\nabla_\perp f(\mathbf{z}_k)}{\|\nabla f(\mathbf{z}_k)\|}\right\| + \frac{\delta}{16L}\sqrt{\frac{\rho\epsilon}{n}}\left\|\hat{\mathbf{g}}_{k,\perp} - \frac{\nabla_\perp f(\mathbf{z}_k)}{\|\nabla f(\mathbf{z}_k)\|}\right\|. \tag{10}$$

Since $\|f(\mathbf{z}_t)\| \geq \frac{\delta r}{16}\sqrt{\frac{\pi\rho\epsilon}{n}}$ by Lemma 11, we have

$$\frac{\delta}{16L}\sqrt{\frac{\rho\epsilon}{n}}\left\|\hat{\mathbf{g}}_{k,\perp} - \frac{\nabla_\perp f(\mathbf{z}_k)}{\|\nabla f(\mathbf{z}_k)\|}\right\| \leq \frac{\delta}{16L}\sqrt{\frac{\rho\epsilon}{n}}\left\|\hat{\mathbf{g}}_k - \frac{\nabla f(\mathbf{z}_k)}{\|\nabla f(\mathbf{z}_k)\|}\right\| \leq \frac{\delta\hat{\delta}}{16L}\sqrt{\frac{\rho\epsilon}{n}} \leq \frac{\delta}{64\mathscr{T}\sqrt{n}}.$$

by Theorem 1. Moreover, observe that

$$\nabla_\perp f(\mathbf{z}_k) = (\nabla^2 f(\mathbf{0})\mathbf{z}_k)_\perp + \nabla_\perp f(\mathbf{0}) + (\nabla_\perp f(\mathbf{z}_k) - \nabla_\perp f(\mathbf{0}) - (\nabla^2 f(\mathbf{0})\mathbf{z}_k)_\perp)$$
$$= \nabla^2 f(\mathbf{0})\mathbf{z}_{k,\perp} + \nabla_\perp f(\mathbf{0}) + (\nabla_\perp f(\mathbf{z}_k) - \nabla_\perp f(\mathbf{0}) - (\nabla^2 f(\mathbf{0})\mathbf{z}_k)_\perp), \tag{11}$$

where the norm of

$$\sigma_{k,\perp} := \nabla_\perp f(\mathbf{z}_k) - \nabla_\perp f(\mathbf{0}) - (\nabla^2 f(\mathbf{0})\mathbf{z}_k)_\perp$$

is upper bounded by

$$\frac{\rho r^2}{2} + \frac{\pi\delta r(\rho\epsilon)^{1/4}\sqrt{L}}{256n\mathscr{T}} \leq \frac{\pi\delta r(\rho\epsilon)^{1/4}\sqrt{L}}{128n\mathscr{T}} \leq \frac{\delta r}{16}\sqrt{\frac{\pi\rho\epsilon}{n}}$$

given that $f$ is $\rho$-Hessian Lipschitz and $\|\nabla f(\mathbf{0})\| \leq \frac{\delta r}{32}\sqrt{\frac{\pi\rho\epsilon}{n}}$. Next, we proceed to bound the first term on the RHS of (10), where

$$\mathbf{y}_{k,\perp} - \frac{\delta}{16L}\sqrt{\frac{\rho\epsilon}{n}}\cdot\frac{\nabla_\perp f(\mathbf{z}_k)}{\|\nabla f(\mathbf{z}_k)\|} = \mathbf{y}_{k,\perp} - \frac{\delta}{16L}\sqrt{\frac{\rho\epsilon}{n}}\cdot\frac{\nabla_\perp f(\mathbf{z}_k)}{\|\nabla f(\mathbf{z}_k)\|}$$
$$= \mathbf{y}_{k,\perp} - \frac{\delta}{16L}\sqrt{\frac{\rho\epsilon}{n}}\cdot\frac{\nabla^2 f(\mathbf{0})\mathbf{z}_{k,\perp}}{\|\nabla f(\mathbf{z}_k)\|} - \frac{\delta}{16L}\sqrt{\frac{\rho\epsilon}{n}}\cdot\frac{\sigma_{k,\perp}}{\|\nabla f(\mathbf{z}_k)\|},$$

where

$$\nabla^2 f(\mathbf{0})\mathbf{z}_{k,\perp} = \sum_{i=2}^n \lambda_i\langle\mathbf{z}_{k,\perp}, \mathbf{u}_i\rangle\mathbf{u}_i = r\sum_{i=2}^n \lambda_i\langle\mathbf{y}_{k,\perp}, \mathbf{u}_i\rangle\mathbf{u}_i,$$

and

$$\mathbf{y}_{k,\perp} - \frac{\delta}{16L}\sqrt{\frac{\rho\epsilon}{n}}\cdot\frac{\nabla^2 f(\mathbf{0})\mathbf{z}_{k,\perp}}{\|\nabla f(\mathbf{z}_k)\|} = \sum_{i=2}^n\left(1 - \frac{r\delta}{16\|\nabla f(\mathbf{z}_k)\|}\sqrt{\frac{\rho\epsilon}{n}}\frac{\lambda_i}{L}\right)\langle\mathbf{y}_{k,\perp}, \mathbf{u}_i\rangle\mathbf{u}_i.$$

Given that

$$-1 \leq \frac{r\delta}{16\|\nabla f(\mathbf{z}_k)\|}\sqrt{\frac{\rho\epsilon}{n}}\frac{\lambda_i}{L} \leq 1$$

is always true, we have

$$\left\| \mathbf{y}_{k,\perp} - \frac{\delta}{16L}\sqrt{\frac{\rho\epsilon}{n}} \cdot \frac{\nabla^2 f(\mathbf{0})\mathbf{z}_{k,\perp}}{\|\nabla f(\mathbf{z}_k)\|} \right\| \leq \left\| \sum_{i=2}^{n}\left(1 + \frac{r\delta\rho\epsilon}{16\|\nabla f(\mathbf{z}_k)\|L\sqrt{n}}\right)\langle \mathbf{y}_{k,\perp}, \mathbf{u}_i\rangle\mathbf{u}_i \right\|$$

$$\leq \left(1 + \frac{r\delta\rho\epsilon}{16\|\nabla f(\mathbf{z}_k)\|L\sqrt{n}}\right)\|\mathbf{y}_{k,\perp}\|$$

and

$$\left\| \mathbf{y}_{k,\perp} - \frac{\delta}{16L}\sqrt{\frac{\rho\epsilon}{n}} \cdot \frac{\nabla_\perp f(\mathbf{z}_k)}{\|\nabla f(\mathbf{z}_k)\|} \right\| \leq \left(1 + \frac{r\delta\rho\epsilon}{16\|\nabla f(\mathbf{z}_k)\|L\sqrt{n}}\right)\|\mathbf{y}_{k,\perp}\| + \left\| \frac{\delta}{16L}\sqrt{\frac{\rho\epsilon}{n}} \cdot \frac{\sigma_{k,\perp}}{\|\nabla f(\mathbf{z}_k)\|} \right\|$$

$$\leq \left(1 + \frac{r\delta\rho\epsilon}{16\|\nabla f(\mathbf{z}_k)\|L\sqrt{n}}\right)\|\mathbf{y}_{k,\perp}\| + \frac{\delta}{64\mathscr{T}\sqrt{n}}.$$

Combined with (10), we can derive that

$$\|\bar{\mathbf{y}}_{k+1,\perp}\| \leq \left\| \mathbf{y}_{k,\perp} - \frac{\delta}{16L}\sqrt{\frac{\rho\epsilon}{n}} \cdot \frac{\nabla_\perp f(\mathbf{z}_k)}{\|\nabla f(\mathbf{z}_k)\|} \right\| + \frac{\delta}{16L}\sqrt{\frac{\rho\epsilon}{n}}\left\| \hat{\mathbf{g}}_{k,\perp} - \frac{\nabla_\perp f(\mathbf{z}_k)}{\|\nabla f(\mathbf{z}_k)\|} \right\| \quad (12)$$

$$\leq \left(1 + \frac{r\delta\rho\epsilon}{16\|\nabla f(\mathbf{z}_k)\|L\sqrt{n}}\right)\|\mathbf{y}_{k,\perp}\| + \frac{\delta}{32\mathscr{T}\sqrt{n}}. \quad (13)$$

Similarly, we have

$$|\bar{y}_{k+1,1}| \geq \left| y_{k,1} - \frac{\delta}{16L}\sqrt{\frac{\rho\epsilon}{n}} \cdot \frac{\nabla_1 f(\mathbf{z}_k)}{\|\nabla f(\mathbf{z}_k)\|} \right| - \frac{\delta}{16L}\sqrt{\frac{\rho\epsilon}{n}}\left| \hat{g}_{k,1} - \frac{\nabla_1 f(\mathbf{z}_k)}{\|\nabla f(\mathbf{z}_k)\|} \right|, \quad (14)$$

where the second term on the RHS of (14) satisfies

$$\frac{\delta}{16L}\sqrt{\frac{\rho\epsilon}{n}}\left| \hat{g}_{k,1} - \frac{\nabla_1 f(\mathbf{z}_k)}{\|\nabla f(\mathbf{z}_k)\|} \right| \leq \frac{\delta}{16L}\sqrt{\frac{\rho\epsilon}{n}}\left\| \hat{\mathbf{g}}_k - \frac{\nabla f(\mathbf{z}_k)}{\|\nabla f(\mathbf{z}_k)\|} \right\| \leq \frac{\delta\hat{\hat{\delta}}}{16L}\sqrt{\frac{\rho\epsilon}{n}} \leq \frac{\delta}{64\mathscr{T}\sqrt{n}},$$

by Theorem 1, whereas the first term on the RHS of (14) satisfies

$$y_{k,1} - \frac{\delta}{16L}\sqrt{\frac{\rho\epsilon}{n}} \cdot \frac{\nabla_1 f(\mathbf{z}_k)}{\|\nabla f(\mathbf{z}_k)\|} = y_{k,1} - \frac{\delta}{16L}\sqrt{\frac{\rho\epsilon}{n}} \cdot \frac{\mathbf{u}_1^\top \nabla^2 f(\mathbf{0})\mathbf{u}_1 y_{k,1}}{\|\nabla f(\mathbf{z}_k)\|} - \frac{\delta}{16L}\sqrt{\frac{\rho\epsilon}{n}} \cdot \frac{\sigma_{k,1}}{\|\nabla f(\mathbf{z}_k)\|}$$

$$= \left(1 + \frac{r\delta\rho\epsilon}{16\|\nabla f(\mathbf{z}_k)\|L\sqrt{n}}\right)y_{k,1} - \frac{\delta}{16L}\sqrt{\frac{\rho\epsilon}{n}} \cdot \frac{\sigma_{k,1}}{\|\nabla f(\mathbf{z}_k)\|},$$

where the absolute value of

$$\sigma_{k,1} := \nabla_1 f(\mathbf{z}_k) - \nabla_1 f(\mathbf{0}) - (\nabla^2 f(\mathbf{0})\mathbf{z}_k)_1$$

is upper bounded by

$$\frac{\rho r^2}{2} + \frac{\pi\delta r(\rho\epsilon)^{1/4}\sqrt{L}}{256n\mathscr{T}} \leq \frac{\pi\delta r(\rho\epsilon)^{1/4}\sqrt{L}}{128n\mathscr{T}} \leq \frac{\delta r}{16}\sqrt{\frac{\pi\rho\epsilon}{n}}$$

given that $f$ is $\rho$-Hessian Lipschitz and

$$\|\nabla f(\mathbf{0})\| \leq \frac{\pi\delta r(\rho\epsilon)^{1/4}\sqrt{L}}{256nT}.$$

Combined with (14), we can derive that

$$|\bar{y}_{k+1,1}| \geq \left| y_{k,1} - \frac{\delta}{16L}\sqrt{\frac{\rho\epsilon}{n}} \cdot \frac{\nabla_1 f(\mathbf{z}_k)}{\|\nabla f(\mathbf{z}_k)\|} \right| - \frac{\delta}{16L}\sqrt{\frac{\rho\epsilon}{n}}\left| \hat{g}_{k,1} - \frac{\nabla_1 f(\mathbf{z}_k)}{\|\nabla f(\mathbf{z}_k)\|} \right|$$

$$\geq \left(1 + \frac{r\delta\rho\epsilon}{16\|\nabla f(\mathbf{z}_k)\|L\sqrt{n}}\right)|y_{k,1}| - \frac{\delta}{32\mathscr{T}\sqrt{n}}.$$

Combined with (12), we have

$$\frac{|y_{k+1,1}|}{\|\mathbf{y}_{k+1,\perp\|}} = \frac{|\bar{y}_{k+1,1}|}{\|\bar{\mathbf{y}}_{k+1,\perp}\|}$$

$$\geq \frac{\left(1 + \frac{r\delta\rho\epsilon}{16\|\nabla f(\mathbf{z}_k)\|L\sqrt{n}}\right)|y_{k,1}| - \frac{\delta}{32\mathscr{T}\sqrt{n}}}{\left(1 + \frac{r\delta\rho\epsilon}{16\|\nabla f(\mathbf{z}_k)\|L\sqrt{n}}\right)\|\mathbf{y}_{k,\perp}\| + \frac{\delta}{32\mathscr{T}\sqrt{n}}}.$$

Hence, if $|y_{k,1}| \geq \frac{1}{2}$, (9) is also true for $t = k+1$. Otherwise, we have $\|\mathbf{y}_{k,\perp}\| \geq \sqrt{3}/2$ and

$$
\frac{|y_{k+1,1}|}{\|\mathbf{y}_{k+1,\perp}\|} \geq \frac{\left(1 + \frac{r\delta\rho\epsilon}{16\|\nabla f(\mathbf{z}_k)\|L\sqrt{n}}\right)|y_{k,1}| - \frac{\delta}{32\mathscr{T}\sqrt{n}}}{\left(1 + \frac{r\delta\rho\epsilon}{16\|\nabla f(\mathbf{z}_k)\|L\sqrt{n}}\right)\|\mathbf{y}_{k,\perp}\| + \frac{\delta}{32\mathscr{T}\sqrt{n}}}
$$

$$
\geq \frac{\left(1 + \frac{r\delta\rho\epsilon}{16\|\nabla f(\mathbf{z}_k)\|L\sqrt{n}} - \frac{1}{8\mathscr{T}}\right)}{\left(1 + \frac{r\delta\rho\epsilon}{16\|\nabla f(\mathbf{z}_k)\|L\sqrt{n}} + \frac{1}{8\mathscr{T}}\right)} \frac{|y_{k,1}|}{\|\mathbf{y}_{k,\perp}\|}
$$

$$
\geq \left(1 - \frac{1}{2\mathscr{T}}\right)\frac{|y_{k,1}|}{\|\mathbf{y}_{k,\perp}\|} \geq \frac{\delta}{2}\sqrt{\frac{\pi}{n}}\left(1 - \frac{1}{2\mathscr{T}}\right)^{k+1}.
$$

Thus, we can conclude that (9) is true for all $t \in [\mathscr{T}]$. This completes the proof. $\qquad\square$

**Lemma 13.** *In the setting of Problem 4, for any $i$ with $\lambda_i \geq -\frac{\sqrt{\rho\epsilon}}{2}$, the $\mathscr{T}$-th iteration of Algorithm 8 satisfies*

$$
\frac{|y_{\mathscr{T},i}|}{|y_{\mathscr{T},1}|} \leq \frac{(\rho\epsilon)^{1/4}}{4\sqrt{nL}} \tag{15}
$$

*if $|y_{0,1}| \geq \frac{\delta}{2}\sqrt{\frac{\pi}{n}}$ and $\|\nabla f(\mathbf{0})\| \leq \frac{\delta r}{32}\sqrt{\frac{\pi\rho\epsilon}{n}}$.*

*Proof.* For any $t \in [\mathscr{T} - 1]$, similar to (14) in the proof of Lemma 12, we have

$$
\bar{y}_{t+1,i} = y_{t,i} - \frac{\delta}{16L}\sqrt{\frac{\rho\epsilon}{n}} \cdot \hat{g}_{t,i},
$$

and

$$
|\bar{y}_{t+1,i}| \leq \left| y_{t,i} - \frac{\delta}{16L}\sqrt{\frac{\rho\epsilon}{n}} \cdot \frac{\nabla_i f(\mathbf{z}_k)}{\|\nabla f(\mathbf{z}_k)\|}\right| - \frac{\delta}{16L}\sqrt{\frac{\rho\epsilon}{n}}\left|\hat{g}_{t,i} - \frac{\nabla_i f(\mathbf{z}_k)}{\|\nabla f(\mathbf{z}_k)\|}\right|. \tag{16}
$$

By Lemma 12 we have $|y_{t,1}| \geq \frac{\delta}{8}\sqrt{\frac{\pi}{n}}$ for each $t \in [\mathscr{T}]$, which combined with Lemma 11 leads to $\|\nabla f(\mathbf{z}_t)\| \geq \frac{\delta r}{16}\sqrt{\frac{\pi\rho\epsilon}{n}}$. Thus, the second term on the RHS of (16) satisfies

$$
\frac{\delta}{16L}\sqrt{\frac{\rho\epsilon}{n}}\left|\hat{g}_{t,i} - \frac{\nabla_i f(\mathbf{z}_t)}{\|\nabla f(\mathbf{z}_t)\|}\right| \leq \frac{\delta}{16L}\sqrt{\frac{\rho\epsilon}{n}}\left\|\hat{\mathbf{g}}_t - \frac{\nabla f(\mathbf{z}_t)}{\|\nabla f(\mathbf{z}_t)\|}\right\| \leq \frac{\delta\hat{\hat{\delta}}}{16L}\sqrt{\frac{\rho\epsilon}{n}} \leq \frac{\delta(\rho\epsilon)^{1/4}}{128\mathscr{T}n}\sqrt{\frac{\pi}{L}}
$$

by Theorem 1. Moreover, the first term on the RHS of (16) satisfies

$$
y_{t,i} - \frac{\delta}{16L}\sqrt{\frac{\rho\epsilon}{n}} \cdot \frac{\nabla_i f(\mathbf{z}_t)}{\|\nabla f(\mathbf{z}_t)\|} = y_{t,i} - \frac{\delta}{16L}\sqrt{\frac{\rho\epsilon}{n}} \cdot \frac{\mathbf{u}_i^\top \nabla^2 f(\mathbf{0})\mathbf{u}_i y_{t,i}}{\|\nabla f(\mathbf{z}_t)\|} - \frac{\delta}{16L}\sqrt{\frac{\rho\epsilon}{n}} \cdot \frac{\sigma_{t,i}}{\|\nabla f(\mathbf{z}_t)\|}
$$

$$
\leq \left(1 + \frac{r\delta\rho\epsilon}{32\|\nabla f(\mathbf{z}_t)\|L\sqrt{n}}\right)y_{t,i} - \frac{\delta}{16L}\sqrt{\frac{\rho\epsilon}{n}} \cdot \frac{\sigma_{t,i}}{\|\nabla f(\mathbf{z}_t)\|},
$$

where the absolute value of

$$
\sigma_{t,i} := \nabla_i f(\mathbf{z}_t) - \nabla_i f(\mathbf{0}) - (\nabla^2 f(\mathbf{0})\mathbf{z}_t)_i
$$

is upper bounded by

$$
\frac{\rho r^2}{2} + \frac{\pi\delta r(\rho\epsilon)^{1/4}\sqrt{L}}{256nT} \leq \frac{\pi\delta r(\rho\epsilon)^{1/4}\sqrt{L}}{128n\mathscr{T}}
$$

given that $f$ is $\rho$-Hessian Lipschitz and

$$
\|\nabla f(\mathbf{0})\| \leq \frac{\pi\delta r(\rho\epsilon)^{1/4}\sqrt{L}}{256n\mathscr{T}}.
$$

Combined with (16), we can derive that

$$
|\bar{y}_{t+1,i}| \leq \left| y_{t,i} - \frac{\delta}{16L}\sqrt{\frac{\rho\epsilon}{n}} \cdot \frac{\nabla_i f(\mathbf{z}_t)}{\|\nabla f(\mathbf{z}_t)\|}\right| + \frac{\delta}{16L}\sqrt{\frac{\rho\epsilon}{n}}\left|\hat{g}_{t,i} - \frac{\nabla_i f(\mathbf{z}_t)}{\|\nabla f(\mathbf{z}_t)\|}\right|
$$

$$
\leq \left(1 + \frac{r\delta\rho\epsilon}{32\|\nabla f(\mathbf{z}_t)\|L\sqrt{n}}\right)|y_{t,i}| + \frac{\delta(\rho\epsilon)^{1/4}}{64\mathscr{T}n}\sqrt{\frac{\pi}{L}}.
$$

Considering that $|y_{t,1}| \geq \frac{\delta}{8}\sqrt{\frac{\pi}{n}}$,

$$
\begin{aligned}
|\bar{y}_{t+1,1}| &\geq \left| y_{t,1} - \frac{\delta}{16L}\sqrt{\frac{\rho\epsilon}{n}} \cdot \frac{\nabla_1 f(\mathbf{z}_t)}{\|\nabla f(\mathbf{z}_t)\|} \right| - \frac{\delta}{16L}\sqrt{\frac{\rho\epsilon}{n}} \left| \hat{g}_{t,1} - \frac{\nabla_1 f(\mathbf{z}_t)}{\|\nabla f(\mathbf{z}_t)\|} \right| \\
&\geq \left( 1 + \frac{r\delta\rho\epsilon}{16\|\nabla f(\mathbf{z}_t)\|L\sqrt{n}} \right)|y_{t,1}| - \frac{\delta(\rho\epsilon)^{1/4}}{64\mathscr{T}n}\sqrt{\frac{\pi}{L}} \\
&\geq \left( 1 + \frac{r\delta\rho\epsilon}{24\|\nabla f(\mathbf{z}_t)\|L\sqrt{n}} \right)|y_{t,1}|,
\end{aligned}
$$

where the last inequality is due to the fact that $|y_{t,1}| \geq \frac{\delta}{8}\sqrt{\frac{\pi}{n}}$ by Lemma 12. Hence, for any $t \in [\mathscr{T}-1]$ we have

$$
\begin{aligned}
\frac{|y_{t+1,i}|}{|y_{t+1,1}|} &= \frac{|\bar{y}_{t+1,i}|}{|\bar{y}_{t+1,1}|} \\
&\leq \frac{\left( 1 + \frac{r\delta\rho\epsilon}{32\|\nabla f(\mathbf{z}_t)\|L\sqrt{n}} \right)|y_{t,i}| + \frac{\delta(\rho\epsilon)^{1/4}}{64\mathscr{T}n}\sqrt{\frac{\pi}{L}}}{\left( 1 + \frac{r\delta\rho\epsilon}{24\|\nabla f(\mathbf{z}_t)\|L\sqrt{n}} \right)|y_{t,1}|} \\
&\leq \frac{\left( 1 + \frac{r\delta\rho\epsilon}{32\|\nabla f(\mathbf{z}_t)\|L\sqrt{n}} \right)|y_{t,i}|}{\left( 1 + \frac{r\delta\rho\epsilon}{24\|\nabla f(\mathbf{z}_t)\|L\sqrt{n}} \right)|y_{t,1}|} + \frac{(\rho\epsilon)^{1/4}}{8\mathscr{T}\sqrt{nL}} \\
&\leq \left( 1 - \frac{r\delta\rho\epsilon}{192\|\nabla f(\mathbf{z}_t)\|L\sqrt{n}} \right)\frac{|y_{t,i}|}{|y_{t,1}|} + \frac{(\rho\epsilon)^{1/4}}{8\mathscr{T}\sqrt{nL}}.
\end{aligned}
$$

Since $f$ is $L$-smooth, we have

$$
\|\nabla f(\mathbf{z}_t)\| \leq \|\nabla f(\mathbf{0})\| + L\|\mathbf{z}_t\| \leq 2Lr,
$$

which leads to

$$
\begin{aligned}
\frac{|y_{t+1,i}|}{|y_{t+1,1}|} &\leq \left( 1 - \frac{r\delta\rho\epsilon}{192\|\nabla f(\mathbf{z}_t)\|L\sqrt{n}} \right)\frac{|y_{t,i}|}{|y_{t,1}|} + \frac{(\rho\epsilon)^{1/4}}{8\mathscr{T}\sqrt{nL}} \\
&\leq \left( 1 - \frac{\delta\rho\epsilon}{384L^2\sqrt{n}} \right)\frac{|y_{t,i}|}{|y_{t,1}|} + \frac{(\rho\epsilon)^{1/4}}{8\mathscr{T}\sqrt{nL}}.
\end{aligned}
$$

Thus,

$$
\begin{aligned}
\frac{|y_{\mathscr{T},i}|}{|y_{\mathscr{T},1}|} &\leq \left( 1 - \frac{\delta\rho\epsilon}{384L^2\sqrt{n}} \right)^{\mathscr{T}} \frac{|y_{0,i}|}{|y_{0,1}|} + \sum_{t=1}^{\mathscr{T}} \frac{(\rho\epsilon)^{1/4}}{6\mathscr{T}\sqrt{nL}} \left( 1 - \frac{\delta\rho\epsilon}{384L^2\sqrt{n}} \right)^{\mathscr{T}-t} \\
&\leq \left( 1 - \frac{\delta\rho\epsilon}{384L^2\sqrt{n}} \right)^{\mathscr{T}} \frac{|y_{0,i}|}{|y_{0,1}|} + \frac{(\rho\epsilon)^{1/4}}{8\sqrt{nL}} \leq \frac{(\rho\epsilon)^{1/4}}{4\sqrt{nL}}.
\end{aligned}
$$

$\square$

Equipped with Lemma 13, we are now ready to prove Lemma 10.

*Proof of Lemma 10.* We consider the case where $|y_{0,1}| \geq \frac{\delta}{2}\sqrt{\frac{\pi}{n}}$, which happens with probability

$$
\Pr\left\{ |y_{0,1}| \geq \frac{\delta}{2}\sqrt{\frac{\pi}{n}} \right\} \geq 1 - \frac{\delta}{2}\sqrt{\frac{\pi}{n}} \cdot \frac{\mathrm{Vol}(\mathcal{S}^{n-2})}{\mathrm{Vol}(\mathcal{S}^{n-1})} \geq 1 - \delta.
$$

In this case, by Lemma 13 we have

$$
|y_{\mathscr{T},1}|^2 = \frac{|y_{\mathscr{T},1}|^2}{\sum_{i=1}^{n}|y_{\mathscr{T},i}|^2} = \left( 1 + \sum_{i=2}^{n}\left( \frac{|y_{\mathscr{T},i}|}{|y_{\mathscr{T},1}|} \right)^2 \right)^{-1} \geq \left( 1 + \frac{\sqrt{\rho\epsilon}}{16L} \right)^{-1} \geq 1 - \frac{\sqrt{\rho\epsilon}}{8L},
$$

and

$$\|\mathbf{y}_{\mathscr{T},\perp}\|^2 = 1 - |y_{\mathscr{T},1}|^2 \le \frac{\sqrt{\rho\epsilon}}{8L}.$$

Let $s$ be the smallest integer such that $\lambda_s \ge 0$. Then the output $\hat{\mathbf{e}} = \mathbf{y}_{\mathscr{T}}$ of Algorithm 8 satisfies

$$\hat{\mathbf{e}}^\top \nabla^2 f(\mathbf{x})\hat{\mathbf{e}} = |y_{\mathscr{T},1}|^2 \mathbf{u}_1^\top \nabla^2 f(\mathbf{x})\mathbf{u}_1 + \mathbf{y}_{\mathscr{T},\perp}^\top \nabla^2 f(\mathbf{x})\mathbf{y}_{\mathscr{T},\perp}$$

$$\le -\sqrt{\rho\epsilon} \cdot |y_{\mathscr{T},1}|^2 + L \sum_{i=s}^{d} \|\mathbf{y}_{\mathscr{T},i}\|^2$$

$$\le -\sqrt{\rho\epsilon} \cdot |y_{\mathscr{T},1}|^2 + L\|\mathbf{y}_{\mathscr{T},\perp}\|^2 \le -\frac{\sqrt{\rho\epsilon}}{4}.$$

The query complexity of Algorithm 8 only comes from the gradient direction estimation step in Line 4, which equals

$$\mathscr{T} \cdot O\left(n\log\left(n/\hat{\delta}\right)\right) = O\left(\frac{L^2 n^{3/2}}{\delta\rho\epsilon} \log^3 \frac{nL}{\delta\sqrt{\rho\epsilon}}\right).$$

$\square$

### C.2.2 NEGATIVE CURVATURE FINDING WHEN THE GRADIENT IS RELATIVELY LARGE

In this part, we present our negative curvature finding algorithm that finds the negative curvature of a point $\mathbf{x}$ with $\lambda_{\min}(\nabla^2 f(\mathbf{x})) \le -\sqrt{\rho\epsilon}$ when the norm of the gradient $\nabla f(\mathbf{x})$ is relatively large.

---

**Algorithm 9:** Comparison-based Negative Curvature Finding 2 (Comparison-NCF2)

**Input:** Function $f\colon \mathbb{R}^n \to \mathbb{R}$, $\mathbf{x}$, precision $\epsilon$, error probability $\delta$

1   $\mathscr{T} \leftarrow \frac{384L^2\sqrt{n}}{\delta\rho\epsilon}\log\frac{36nL}{\sqrt{\rho\epsilon}}, \hat{\delta} \leftarrow \frac{1}{8\mathscr{T}(\rho\epsilon)^{1/4}}\sqrt{\frac{\pi L}{n}}, \gamma_{\mathbf{x}} \leftarrow \frac{\pi\delta r(\rho\epsilon)^{1/4}\sqrt{L}}{256n\mathscr{T}}, \gamma_{\mathbf{y}} \leftarrow \frac{\delta}{8}\sqrt{\frac{\pi}{n}}$

2   $\mathbf{y}_0 \leftarrow \text{Uniform}(\mathcal{S}^{n-1})$

3   **for** $t = 0,\ldots,\mathscr{T}-1$ **do**

4      $\hat{\mathbf{g}}_t \leftarrow \text{Comparison-Hessian-Vector}(\mathbf{x},\mathbf{y}_t,\hat{\delta},\gamma_{\mathbf{x}},\gamma_{\mathbf{y}})$

5      $\bar{\mathbf{y}}_{t+1} \leftarrow \mathbf{y}_t - \frac{\delta}{16L}\sqrt{\frac{\rho\epsilon}{n}}\hat{\mathbf{g}}_t$

6      $\mathbf{y}_{t+1} \leftarrow \bar{\mathbf{y}}_{t+1}/\|\bar{\mathbf{y}}_{t+1}\|$

7   **return** $\hat{\mathbf{e}} \leftarrow \mathbf{y}_{\mathscr{T}}$

---

The subroutine `Comparison-Hessian-Vector` in Line 4 of Algorithm 9 is given as Algorithm 10, whose output approximates the Hessian-vector product $\nabla^2 f(\mathbf{x}) \cdot \mathbf{y}_t$.

---

**Algorithm 10:** Comparison-based Hessian-vector product (Comparison-Hessian-Vector)

**Input:** Function $f\colon \mathbb{R}^n \to \mathbb{R}$, $\mathbf{x},\mathbf{y} \in \mathbb{R}^n$, precision $\hat{\delta}$, lower bound $\gamma_{\mathbf{x}}$ on $\|\nabla f(\mathbf{x})\|$, lower bound $\gamma_{\mathbf{y}}$ on $|y_1|$

1   Set $r_0 \leftarrow \min\left\{\frac{\gamma_{\mathbf{x}}}{100L}, \frac{\gamma_{\mathbf{x}}}{100\rho}, \frac{\sqrt{\gamma_{\mathbf{x}}\hat{\delta}}}{20\sqrt{\rho}}, \frac{\gamma_{\mathbf{y}}\hat{\delta}\sqrt{\epsilon}}{20\sqrt{\rho}}\right\}$

2   $\hat{\mathbf{g}}_0 \leftarrow \text{Comparison-GDE}(\mathbf{x}, \frac{\rho r_0^2}{\gamma_{\mathbf{x}}}, \gamma_{\mathbf{x}}), \hat{\mathbf{g}}_1 \leftarrow \text{Comparison-GDE}(\mathbf{x}+r_0\mathbf{y}, \frac{\rho r_0^2}{\gamma_{\mathbf{x}}}, \gamma_{\mathbf{x}}/2),$

     $\hat{\mathbf{g}}_{-1} \leftarrow \text{Comparison-GDE}(\mathbf{x}-r_0\mathbf{y}, \frac{\rho r_0^2}{\gamma_{\mathbf{x}}}, \gamma_{\mathbf{x}}/2)$

3   Set $\mathbf{g} = \sqrt{1-\langle\hat{\mathbf{g}}_{-1},\hat{\mathbf{g}}_0\rangle^2}\hat{\mathbf{g}}_1 - \sqrt{1-\langle\hat{\mathbf{g}}_1,\hat{\mathbf{g}}_0\rangle^2}\hat{\mathbf{g}}_{-1}$

4   **return** $\hat{\mathbf{g}} = \mathbf{g}/\|\mathbf{g}\|$

---

**Lemma 14.** *In the setting of Problem 4, for any $\mathbf{x},\mathbf{y} \in \mathbb{R}^d$ satisfying*

$$\|\nabla f(\mathbf{x})\| \ge \gamma_{\mathbf{x}}, \quad \lambda_{\min}(\nabla^2 f(\mathbf{x})) \le -\sqrt{\rho\epsilon}, \quad \|\mathbf{y}\| = 1, \quad |y_1| \ge \gamma_{\mathbf{y}},$$

*Algorithm 10 outputs a vector $\hat{\mathbf{g}}$ satisfying*

$$\left\|\hat{\mathbf{g}} - \frac{\nabla^2 f(\mathbf{x}) \cdot \mathbf{y}}{\|\nabla^2 f(\mathbf{x}) \cdot \mathbf{y}\|}\right\| \le \hat{\delta}$$

*using $O\left(n\log\left(n\rho L^2/\gamma_{\mathbf{x}}\gamma_{\mathbf{y}}^2\epsilon\hat{\delta}^2\right)\right)$ queries.*

*Proof of Lemma 14.* Since $f$ is a $\rho$-Hessian Lipschitz function,

$$\left\|\nabla f(\mathbf{x} + r_0\mathbf{y}) - \nabla f(\mathbf{x}) - r_0\nabla^2 f(\mathbf{x}) \cdot \mathbf{y}\right\| \le \frac{\rho}{2}r_0^2; \tag{17}$$

$$\left\|\nabla f(\mathbf{x} - r_0\mathbf{y}) - \nabla f(\mathbf{x}) + r_0\nabla^2 f(\mathbf{x}) \cdot \mathbf{y}\right\| \le \frac{\rho}{2}r_0^2. \tag{18}$$

Therefore,

$$\left\|\nabla f(\mathbf{x} + r_0\mathbf{y}) + \nabla f(\mathbf{x} - r_0\mathbf{y}) - 2\nabla f(\mathbf{x})\right\| \le \rho r_0^2; \tag{19}$$

$$\left\|\nabla^2 f(\mathbf{x}) \cdot \mathbf{y} - \frac{1}{2r_0}\left(\nabla f(\mathbf{x} + r_0\mathbf{y}) - \nabla f(\mathbf{x} - r_0\mathbf{y})\right)\right\| \le \frac{\rho}{2}r_0. \tag{20}$$

Furthermore, because $r_0 \le \frac{\gamma_{\mathbf{x}}}{100L}$ and $f$ is $L$-smooth,

$$\|\nabla f(\mathbf{x} + r_0\mathbf{y})\|, \|\nabla f(\mathbf{x} - r_0\mathbf{y})\| \ge \gamma_{\mathbf{x}} - L \cdot \frac{\gamma_{\mathbf{x}}}{100L} = 0.99\gamma_{\mathbf{x}}.$$

We first understand how to approximate $\nabla^2 f(\mathbf{x}) \cdot \mathbf{y}$ by normalized vectors $\frac{\nabla f(\mathbf{x})}{\|\nabla f(\mathbf{x})\|}, \frac{\nabla f(\mathbf{x}+r_0\mathbf{y})}{\|\nabla f(\mathbf{x}+r_0\mathbf{y})\|}, \frac{\nabla f(\mathbf{x}-r_0\mathbf{y})}{\|\nabla f(\mathbf{x}-r_0\mathbf{y})\|}$, and then analyze the approximation error due to using $\hat{\mathbf{g}}_0, \hat{\mathbf{g}}_1, \hat{\mathbf{g}}_{-1}$, respectively. By Lemma 7, we have

$$\frac{1}{2\|\nabla f(\mathbf{x})\|} \frac{\|\nabla f(\mathbf{x}) - r_0\nabla^2 f(\mathbf{x}) \cdot \mathbf{y}\|}{\sqrt{1 - \left\langle \frac{\nabla f(\mathbf{x})+r_0\nabla^2 f(\mathbf{x})\cdot\mathbf{y}}{\|\nabla f(\mathbf{x})+r_0\nabla^2 f(\mathbf{x})\cdot\mathbf{y}\|}, \frac{\nabla f(\mathbf{x})}{\|\nabla f(\mathbf{x})\|} \right\rangle^2}}$$

$$= \frac{1}{2\|\nabla f(\mathbf{x})\|} \frac{\|\nabla f(\mathbf{x}) + r_0\nabla^2 f(\mathbf{x}) \cdot \mathbf{y}\|}{\sqrt{1 - \left\langle \frac{\nabla f(\mathbf{x})-r_0\nabla^2 f(\mathbf{x})\cdot\mathbf{y}}{\|\nabla f(\mathbf{x})-r_0\nabla^2 f(x)\cdot\mathbf{y}\|}, \frac{\nabla f(\mathbf{x})}{\|\nabla f(\mathbf{x})\|} \right\rangle^2}} =: \alpha, \tag{21}$$

i.e., we denote the value above as $\alpha$. Because $f$ is $\rho$-Hessian Lipschitz, $\|r_0\nabla^2 f(\mathbf{x}) \cdot \mathbf{y}\| \le r_0\rho$. Since $r_0 \le \frac{\gamma_{\mathbf{x}}}{100\rho}$, $\|r_0\nabla^2 f(\mathbf{x}) \cdot \mathbf{y}\| \le \frac{\gamma_{\mathbf{x}}}{100}$. Also note that by Lemma 6 we have

$$\left\langle \frac{\nabla f(\mathbf{x}) + r_0\nabla^2 f(\mathbf{x}) \cdot \mathbf{y}}{\|\nabla f(\mathbf{x}) + r_0\nabla^2 f(\mathbf{x}) \cdot \mathbf{y}\|}, \frac{\nabla f(\mathbf{x})}{\|\nabla f(\mathbf{x})\|} \right\rangle \ge 0.94, \quad \left\langle \frac{\nabla f(\mathbf{x}) - r_0\nabla^2 f(\mathbf{x}) \cdot \mathbf{y}}{\|\nabla f(\mathbf{x}) - r_0\nabla^2 f(\mathbf{x}) \cdot \mathbf{y}\|}, \frac{\nabla f(\mathbf{x})}{\|\nabla f(\mathbf{x})\|} \right\rangle \ge 0.94.$$

This promises that

$$\alpha \ge \frac{0.99}{2\sqrt{1 - 0.94^2}} \ge 1. \tag{22}$$

In arguments next, we say a vector $\mathbf{u}$ is $d$-close to a vector $\mathbf{v}$ if $\|\mathbf{u} - \mathbf{v}\| \le d$. We prove that the vector

$$\tilde{\mathbf{g}}_1 := \frac{\nabla f(\mathbf{x})}{\|\nabla f(\mathbf{x})\|} + \alpha \cdot \left( \sqrt{1 - \left\langle \frac{\nabla f(\mathbf{x} - r_0\mathbf{y})}{\|\nabla f(\mathbf{x} - r_0\mathbf{y})\|}, \frac{\nabla f(\mathbf{x})}{\|\nabla f(\mathbf{x})\|} \right\rangle^2} \frac{\nabla f(\mathbf{x} + r_0\mathbf{y})}{\|\nabla f(\mathbf{x} + r_0\mathbf{y})\|} \right.$$

$$\left. - \sqrt{1 - \left\langle \frac{\nabla f(\mathbf{x} + r_0\mathbf{y})}{\|\nabla f(\mathbf{x} + r_0\mathbf{y})\|}, \frac{\nabla f(\mathbf{x})}{\|\nabla f(\mathbf{x})\|} \right\rangle^2} \frac{\nabla f(\mathbf{x} - r_0\mathbf{y})}{\|\nabla f(\mathbf{x} - r_0\mathbf{y})\|} \right) \tag{23}$$

is $\frac{7\rho r_0^2}{\gamma_{\mathbf{x}}}$-close to a vector proportional to $\nabla f(\mathbf{x} + r_0\mathbf{y})$. This is because (17), (18), and Lemma 5 imply that

$$\frac{\nabla f(\mathbf{x} + r_0\mathbf{y})}{\|\nabla f(\mathbf{x} + r_0\mathbf{y})\|} \quad \text{and} \quad \frac{\nabla f(\mathbf{x}) + r_0\nabla^2 f(\mathbf{x}) \cdot \mathbf{y}}{\|\nabla f(\mathbf{x}) + r_0\nabla^2 f(\mathbf{x}) \cdot \mathbf{y}\|}$$

are $\frac{\rho r_0^2}{0.99\gamma_{\mathbf{x}}}$-close to each other,

$$\sqrt{1 - \left\langle \frac{\nabla f(\mathbf{x} - r_0\mathbf{y})}{\|\nabla f(\mathbf{x} - r_0\mathbf{y})\|}, \frac{\nabla f(\mathbf{x})}{\|\nabla f(\mathbf{x})\|} \right\rangle^2} \frac{\nabla f(\mathbf{x} + r_0\mathbf{y})}{\|\nabla f(\mathbf{x} + r_0\mathbf{y})\|} \tag{24}$$

is proportional to $\nabla f(\mathbf{x} + r_0 \mathbf{y})$, and the definition of $\alpha$ implies

$$\frac{\nabla f(\mathbf{x})}{\|\nabla f(\mathbf{x})\|} - \alpha \sqrt{1 - \left\langle \frac{\nabla f(\mathbf{x}) + r_0 \nabla^2 f(\mathbf{x}) \cdot \mathbf{y}}{\|\nabla f(\mathbf{x}) + r_0 \nabla^2 f(\mathbf{x}) \cdot \mathbf{y}\|}, \frac{\nabla f(\mathbf{x})}{\|\nabla f(\mathbf{x})\|} \right\rangle^2} \frac{\nabla f(\mathbf{x}) - r_0 \nabla^2 f(\mathbf{x}) \cdot \mathbf{y}}{\|\nabla f(\mathbf{x}) - r_0 \nabla^2 f(\mathbf{x}) \cdot \mathbf{y}\|}$$
$$= \frac{\nabla f(\mathbf{x}) + r_0 \nabla^2 f(\mathbf{x}) \cdot \mathbf{y}}{2\|\nabla f(\mathbf{x})\|}. \tag{25}$$

The above vector is $\frac{\rho r_0^2}{4\gamma_{\mathbf{x}}}$-close to $\frac{\nabla f(\mathbf{x}+r_0\mathbf{y})}{2\|\nabla f(\mathbf{x})\|}$ by (17), and the error in above steps cumulates by at most $\frac{6\rho r_0^2}{0.99\gamma_{\mathbf{x}}}$ using Lemma 6. In total $\frac{6\rho r_0^2}{0.99\gamma_{\mathbf{x}}} + \frac{\rho r_0^2}{4\gamma_{\mathbf{x}}} \le \frac{7\rho r_0^2}{\gamma_{\mathbf{x}}}$.

Furthermore, this vector proportional to $\nabla f(\mathbf{x} + r_0 \mathbf{y})$ that is $\frac{\rho r_0^2}{4\gamma_{\mathbf{x}}}$-close to (23) has norm at least $(1 - 0.01)/2 = 0.495$ because the coefficient in (24) is positive, while in the equality above we have $\|r_0 \nabla^2 f(\mathbf{x}) \cdot \mathbf{y}\| \le \frac{\gamma_{\mathbf{x}}}{100}$. Therefore, applying Lemma 5, the vector $\tilde{\mathbf{g}}_1$ in (23) satisfies

$$\left\| \frac{\tilde{\mathbf{g}}_1}{\|\tilde{\mathbf{g}}_1\|} - \frac{\nabla f(\mathbf{x} + r_0 \mathbf{y})}{\|\nabla f(\mathbf{x} + r_0 \mathbf{y})\|} \right\| \le \frac{29\rho r_0^2}{\gamma_{\mathbf{x}}}. \tag{26}$$

Following the same proof, we can prove that the vector

$$\tilde{\mathbf{g}}_{-1} := \frac{\nabla f(\mathbf{x})}{\|\nabla f(\mathbf{x})\|} - \alpha \cdot \left( \sqrt{1 - \left\langle \frac{\nabla f(\mathbf{x} - r_0 \mathbf{y})}{\|\nabla f(\mathbf{x} - r_0 \mathbf{y})\|}, \frac{\nabla f(\mathbf{x})}{\|\nabla f(\mathbf{x})\|} \right\rangle^2} \frac{\nabla f(\mathbf{x} + r_0 \mathbf{y})}{\|\nabla f(\mathbf{x} + r_0 \mathbf{y})\|} \right.$$
$$\left. - \sqrt{1 - \left\langle \frac{\nabla f(\mathbf{x} + r_0 \mathbf{y})}{\|\nabla f(\mathbf{x} + r_0 \mathbf{y})\|}, \frac{\nabla f(\mathbf{x})}{\|\nabla f(\mathbf{x})\|} \right\rangle^2} \frac{\nabla f(\mathbf{x} - r_0 \mathbf{y})}{\|\nabla f(\mathbf{x} - r_0 \mathbf{y})\|} \right) \tag{27}$$

satisfies

$$\left\| \frac{\tilde{\mathbf{g}}_{-1}}{\|\tilde{\mathbf{g}}_{-1}\|} - \frac{\nabla f(\mathbf{x} - r_0 \mathbf{y})}{\|\nabla f(\mathbf{x} - r_0 \mathbf{y})\|} \right\| \le \frac{29\rho r_0^2}{\gamma_{\mathbf{x}}}. \tag{28}$$

Furthermore, (25) implies that $\tilde{\mathbf{g}}_1 - \tilde{\mathbf{g}}_{-1}$ is $2 \cdot \frac{7\rho r_0^2}{\gamma_{\mathbf{x}}} = \frac{14\rho r_0^2}{\gamma_{\mathbf{x}}}$-close to

$$\frac{\nabla f(\mathbf{x}) + r_0 \nabla^2 f(\mathbf{x}) \cdot \mathbf{y}}{2\|\nabla f(\mathbf{x})\|} - \frac{\nabla f(\mathbf{x}) - r_0 \nabla^2 f(\mathbf{x}) \cdot \mathbf{y}}{2\|\nabla f(\mathbf{x})\|} = \frac{r_0}{\|\nabla f(\mathbf{x})\|} \nabla^2 f(\mathbf{x}) \cdot \mathbf{y}. \tag{29}$$

Because $\lambda_{\min}(\nabla^2 f(\mathbf{x})) \le -\sqrt{\rho \epsilon}$ and $|y_1| \ge \gamma_{\mathbf{y}}$, $\|\nabla^2 f(\mathbf{x}) \cdot \mathbf{y}\| \ge \sqrt{\rho \epsilon} \gamma_{\mathbf{y}}$. Therefore, the RHS of (29) has norm at least $\frac{r_0 \sqrt{\rho \epsilon} \gamma_{\mathbf{y}}}{\gamma_{\mathbf{x}}}$, and by Lemma 5 we have

$$\left\| \frac{\tilde{\mathbf{g}}_1 - \tilde{\mathbf{g}}_{-1}}{\|\tilde{\mathbf{g}}_1 - \tilde{\mathbf{g}}_{-1}\|} - \frac{\nabla^2 f(\mathbf{x}) \cdot \mathbf{y}}{\|\nabla^2 f(\mathbf{x}) \cdot \mathbf{y}\|} \right\| \le \frac{14\rho r_0^2}{\gamma_{\mathbf{x}}} / \frac{r_0 \sqrt{\rho \epsilon} \gamma_{\mathbf{y}}}{\gamma_{\mathbf{x}}} = \frac{14 r_0 \sqrt{\rho}}{\sqrt{\epsilon} \gamma_{\mathbf{y}}}. \tag{30}$$

Finally, by Theorem 1 and our choice of the precision parameter, the error coming from running `Comparison-GDE` is:

$$\left\| \hat{\mathbf{g}}_0 - \frac{\nabla f(\mathbf{x})}{\|\nabla f(\mathbf{x})\|} \right\|, \left\| \hat{\mathbf{g}}_1 - \frac{\nabla f(\mathbf{x} + r_0 \mathbf{y})}{\|\nabla f(\mathbf{x} + r_0 \mathbf{y})\|} \right\|, \left\| \hat{\mathbf{g}}_{-1} - \frac{\nabla f(\mathbf{x} - r_0 \mathbf{y})}{\|\nabla f(\mathbf{x} - r_0 \mathbf{y})\|} \right\| \le \frac{\rho r_0^2}{\gamma_{\mathbf{x}}}. \tag{31}$$

Combined with (26) and (28), we know that the vector $\mathbf{g}$ we obtained in Algorithm 10 is

$$\frac{29\rho r_0^2}{\gamma_{\mathbf{x}}} + \frac{29\rho r_0^2}{\gamma_{\mathbf{x}}} + 3 \cdot \frac{\rho r_0^2}{\gamma_{\mathbf{x}}} = \frac{61\rho r_0^2}{\gamma_{\mathbf{x}}} \tag{32}$$

close to $(\tilde{\mathbf{g}}_1 - \tilde{\mathbf{g}}_{-1})/2\alpha$. Since $\alpha \ge 1$ by (22), by Lemma 5 we have

$$\left\| \frac{\mathbf{g}}{\|\mathbf{g}\|} - \frac{\tilde{\mathbf{g}}_1 - \tilde{\mathbf{g}}_{-1}}{\|\tilde{\mathbf{g}}_1 - \tilde{\mathbf{g}}_{-1}\|} \right\| \le \frac{61\rho r_0^2}{\gamma_{\mathbf{x}}}. \tag{33}$$

In total, all the errors we have accumulated are (30) and (33):

$$\left\| \frac{\mathbf{g}}{\|\mathbf{g}\|} - \frac{\nabla^2 f(\mathbf{x}) \cdot \mathbf{y}}{\|\nabla^2 f(\mathbf{x}) \cdot \mathbf{y}\|} \right\| \leq \frac{61\rho r_0^2}{\gamma_{\mathbf{x}}} + \frac{14 r_0 \sqrt{\rho}}{\sqrt{\epsilon}\gamma_{\mathbf{y}}}. \tag{34}$$

Our selection of $r_0 = \min\left\{ \frac{\gamma_{\mathbf{x}}}{100L}, \frac{\gamma_{\mathbf{x}}}{100\rho}, \frac{\sqrt{\gamma_{\mathbf{x}}\hat{\delta}}}{20\sqrt{\rho}}, \frac{\gamma_{\mathbf{y}}\hat{\delta}\sqrt{\epsilon}}{20\sqrt{\rho}} \right\}$ can guarantee that (34) is at most $\hat{\delta}$.

In terms of query complexity, we made 3 calls to `Comparison-GDE`. By Theorem 1 and that our precision is

$$\frac{\rho r_0^2}{\gamma_{\mathbf{x}}} = \Omega\left( \frac{\gamma_{\mathbf{x}}\gamma_{\mathbf{y}}^2 \epsilon \hat{\delta}^2}{\rho L^2} \right),$$

the total query complexity is $O\left( n \log\left( n\rho L^2 / \gamma_{\mathbf{x}}\gamma_{\mathbf{y}}^2 \epsilon \hat{\delta}^2 \right) \right)$.  □

Based on Lemma 14, we obtain the following result.

**Lemma 15.** *In the setting of Problem 4, for any $\mathbf{x}$ satisfying*

$$\|\nabla f(\mathbf{x})\| \geq L \left( \frac{\pi\delta}{256 n \mathscr{T}} \right)^2 \sqrt{\frac{\epsilon}{\rho}}, \qquad \lambda_{\min}(\nabla^2 f(\mathbf{x})) \leq -\sqrt{\rho\epsilon},$$

*Algorithm 9 outputs a unit vector $\hat{\mathbf{e}}$ satisfying*

$$\hat{\mathbf{e}}^\top \nabla^2 f(\mathbf{x}) \hat{\mathbf{e}} \leq -\sqrt{\rho\epsilon}/4,$$

*with success probability at least $1 - \delta$ using*

$$O\left( \frac{L^2 n^{3/2}}{\delta\rho\epsilon} \log^2 \frac{nL}{\delta\sqrt{\rho\epsilon}} \right)$$

*queries.*

The proof of Lemma 15 is similar to the proof of Lemma 10. Without loss of generality we assume $\mathbf{x} = \mathbf{0}$ by shifting $\mathbb{R}^n$ such that $\mathbf{x}$ is mapped to $\mathbf{0}$. We denote $\mathbf{g}_t := \nabla^2 f(\mathbf{0}) \cdot \mathbf{y}_t$ for each iteration $t \in [\mathscr{T}]$ of Algorithm 9.

**Lemma 16.** *In the setting of Problem 4, for any iteration $t \in [\mathscr{T}]$ of Algorithm 9 we have*

$$|y_{t,1}| \geq \frac{\delta}{8}\sqrt{\frac{\pi}{n}} \tag{35}$$

*if $|y_{0,1}| \geq \frac{\delta}{2}\sqrt{\frac{\pi}{n}}$ and $\|\nabla f(\mathbf{0})\| \leq \frac{\delta r}{32}\sqrt{\frac{\pi\rho\epsilon}{n}}$.*

*Proof.* We use recurrence to prove this lemma. In particular, assume

$$\frac{|y_{t,1}|}{\|\mathbf{y}_{t,\perp}\|} \geq \frac{\delta}{2}\sqrt{\frac{\pi}{n}} \left( 1 - \frac{1}{2\mathscr{T}} \right)^t \tag{36}$$

is true for all $t \leq k$ for some $k$, which guarantees that

$$|y_{t,1}| \geq \frac{\delta}{4}\sqrt{\frac{\pi}{n}} \left( 1 - \frac{1}{2\mathscr{T}} \right)^t$$

Then for $t = k + 1$, we have

$$\bar{\mathbf{y}}_{k+1,\perp} = \mathbf{y}_{k,\perp} - \frac{\delta}{16L}\sqrt{\frac{\rho\epsilon}{n}} \cdot \hat{\mathbf{g}}_{k,\perp},$$

and

$$\|\bar{\mathbf{y}}_{k+1,\perp}\| \leq \left\| \mathbf{y}_{k,\perp} - \frac{\delta}{16L}\sqrt{\frac{\rho\epsilon}{n}} \cdot \frac{\mathbf{g}_{k,\perp}}{\|\mathbf{g}_k\|} \right\| + \frac{\delta}{16L}\sqrt{\frac{\rho\epsilon}{n}} \left\| \hat{\mathbf{g}}_{k,\perp} - \frac{\mathbf{g}_{k,\perp}}{\|\mathbf{g}_k\|} \right\|, \tag{37}$$

where

$$\frac{\delta}{16L}\sqrt{\frac{\rho\epsilon}{n}}\left\|\hat{\mathbf{g}}_{k,\perp}-\frac{\mathbf{g}_{k,\perp}}{\|\mathbf{g}_k\|}\right\| \le \frac{\delta}{16L}\sqrt{\frac{\rho\epsilon}{n}}\left\|\hat{\mathbf{g}}_k-\frac{\mathbf{g}_k}{\|\mathbf{g}_k\|}\right\| \le \frac{\delta\hat{\delta}}{16L}\sqrt{\frac{\rho\epsilon}{n}} \le \frac{\delta}{64\mathscr{T}\sqrt{n}}.$$

by Lemma 14. Next, we proceed to bound the first term on the RHS of (37). Note that

$$\mathbf{g}_{k,\perp} = \nabla^2 f(\mathbf{0})\mathbf{y}_{k,\perp} = \sum_{i=2}^{n}\lambda_i\langle\mathbf{y}_{k,\perp},\mathbf{u}_i\rangle\mathbf{u}_i,$$

and

$$\mathbf{y}_{k,\perp} - \frac{\delta}{16L}\sqrt{\frac{\rho\epsilon}{n}}\cdot\frac{\mathbf{g}_{k,\perp}}{\|\mathbf{g}_k\|} = \sum_{i=2}^{n}\left(1-\frac{\delta}{16\|\mathbf{g}_k\|}\sqrt{\frac{\rho\epsilon}{n}}\frac{\lambda_i}{L}\right)\langle\mathbf{y}_{k,\perp},\mathbf{u}_i\rangle\mathbf{u}_i,$$

where

$$\|\mathbf{g}_k\| \ge |g_{k,1}| \ge \sqrt{\rho\epsilon}|y_{k,1}| \ge \frac{\delta}{8}\sqrt{\frac{\pi}{n}}.$$

Consequently, we have

$$-1 \le \frac{\delta}{16\|\mathbf{g}_k\|}\sqrt{\frac{\rho\epsilon}{n}}\frac{\lambda_i}{L} \le 1, \qquad \forall i = 1,\dots,n,$$

which leads to

$$\left\|\mathbf{y}_{k,\perp}-\frac{\delta}{16L}\sqrt{\frac{\rho\epsilon}{n}}\cdot\frac{\mathbf{g}_{k,\perp}}{\|\mathbf{g}_k\|}\right\| \le \left\|\sum_{i=2}^{n}\left(1+\frac{\delta\rho\epsilon}{16\|\mathbf{g}_k\|L\sqrt{n}}\right)\langle\mathbf{y}_{k,\perp},\mathbf{u}_i\rangle\mathbf{u}_i\right\|$$

$$\le \left(1+\frac{\delta\rho\epsilon}{16\|\mathbf{g}_k\|L\sqrt{n}}\right)\|\mathbf{y}_{k,\perp}\|.$$

Combined with (37), we can derive that

$$\|\bar{\mathbf{y}}_{k+1,\perp}\| \le \left\|\mathbf{y}_{k,\perp}-\frac{\delta}{16L}\sqrt{\frac{\rho\epsilon}{n}}\cdot\frac{\mathbf{g}_{k,\perp}}{\|\mathbf{g}_k\|}\right\| + \frac{\delta}{16L}\sqrt{\frac{\rho\epsilon}{n}}\left\|\hat{\mathbf{g}}_{k,\perp}-\frac{\mathbf{g}_{k,\perp}}{\|\mathbf{g}_k\|}\right\| \tag{38}$$

$$\le \left(1+\frac{\delta\rho\epsilon}{16\|\mathbf{g}_k\|L\sqrt{n}}\right)\|\mathbf{y}_{k,\perp}\| + \frac{\delta}{64\mathscr{T}\sqrt{n}}. \tag{39}$$

Similarly, we have

$$|\bar{y}_{k+1,1}| \ge \left|y_{k,1}-\frac{\delta}{16L}\sqrt{\frac{\rho\epsilon}{n}}\cdot\frac{g_{k,1}}{\|\mathbf{g}_k\|}\right| - \frac{\delta}{16L}\sqrt{\frac{\rho\epsilon}{n}}\left|\hat{g}_{k,1}-\frac{g_{k,1}}{\|\mathbf{g}_k\|}\right|, \tag{40}$$

where the second term on the RHS of (40) satisfies

$$\frac{\delta}{16L}\sqrt{\frac{\rho\epsilon}{n}}\left|\hat{g}_{k,1}-\frac{g_{k,1}}{\|\mathbf{g}_k\|}\right| \le \frac{\delta}{16L}\sqrt{\frac{\rho\epsilon}{n}}\left\|\hat{\mathbf{g}}_k-\frac{\mathbf{g}_k}{\|\mathbf{g}_k\|}\right\| \le \frac{\delta\hat{\delta}}{16L}\sqrt{\frac{\rho\epsilon}{n}} \le \frac{\delta}{64\mathscr{T}\sqrt{n}},$$

by Lemma 14. Combined with (40), we can derive that

$$|\bar{y}_{k+1,1}| \ge \left|y_{k,1}-\frac{\delta}{16L}\sqrt{\frac{\rho\epsilon}{n}}\cdot\frac{g_{k,1}}{\|\mathbf{g}_k\|}\right| - \frac{\delta}{16L}\sqrt{\frac{\rho\epsilon}{n}}\left|\hat{g}_{k,1}-\frac{g_{k,1}}{\|\mathbf{g}_k\|}\right|$$

$$\ge \left(1+\frac{\delta\rho\epsilon}{16\|\mathbf{g}_k\|L\sqrt{n}}\right)|y_{k,1}| - \frac{\delta}{64\mathscr{T}\sqrt{n}}.$$

Consequently,

$$\frac{|y_{k+1,1}|}{\|\mathbf{y}_{k+1,\perp}\|} = \frac{|\bar{y}_{k+1,1}|}{\|\bar{\mathbf{y}}_{k+1,\perp}\|}$$

$$\ge \frac{\left(1+\frac{\delta\rho\epsilon}{16\|\mathbf{g}_k\|L\sqrt{n}}\right)|y_{k,1}| - \frac{\delta}{64\mathscr{T}\sqrt{n}}}{\left(1+\frac{\delta\rho\epsilon}{16\|\mathbf{g}_k\|L\sqrt{n}}\right)\|\mathbf{y}_{k,\perp}\| + \frac{\delta}{64\mathscr{T}\sqrt{n}}}.$$

Thus, if $|y_{k,1}| \geq \frac{1}{2}$, (36) is also true for $t = k+1$. Otherwise, we have $\|\mathbf{y}_{k,\perp}\| \geq \sqrt{3}/2$ and

$$
\begin{aligned}
\frac{|y_{k+1,1}|}{\|\mathbf{y}_{k+1,\perp}\|} &\geq \frac{\left(1 + \frac{\delta\rho\epsilon}{16\|\nabla f(\mathbf{z}_k)\|L\sqrt{n}}\right)|y_{k,1}| - \frac{\delta}{64\mathscr{T}\sqrt{n}}}{\left(1 + \frac{\delta\rho\epsilon}{16\|\mathbf{g}_k\|L\sqrt{n}}\right)\|\mathbf{y}_{k,\perp}\| + \frac{\delta}{64\mathscr{T}\sqrt{n}}} \\
&\geq \frac{\left(1 + \frac{\delta\rho\epsilon}{16\|\mathbf{g}_k\|L\sqrt{n}} - \frac{1}{8\mathscr{T}}\right)}{\left(1 + \frac{\delta\rho\epsilon}{16\|\mathbf{g}_k\|L\sqrt{n}} + \frac{1}{8\mathscr{T}}\right)} \frac{|y_{k,1}|}{\|\mathbf{y}_{k,\perp}\|} \\
&\geq \left(1 - \frac{1}{2\mathscr{T}}\right)\frac{|y_{k,1}|}{\|\mathbf{y}_{k,\perp}\|} \geq \frac{\delta}{2}\sqrt{\frac{\pi}{n}}\left(1 - \frac{1}{2\mathscr{T}}\right)^{k+1}.
\end{aligned}
$$

Thus, we can conclude that (36) is true for all $t \in [\mathscr{T}]$. This completes the proof. $\qquad\square$

**Lemma 17.** *In the setting of Problem 4, for any $i$ with $\lambda_i \geq -\frac{\sqrt{\rho\epsilon}}{2}$, the $\mathscr{T}$-th iteration of Algorithm 9 satisfies*

$$
\frac{|y_{\mathscr{T},i}|}{|y_{\mathscr{T},1}|} \leq \frac{(\rho\epsilon)^{1/4}}{4\sqrt{nL}} \tag{41}
$$

*if $|y_{0,1}| \geq \frac{\delta}{2}\sqrt{\frac{\pi}{n}}$.*

*Proof.* For any $t \in [\mathscr{T} - 1]$, similar to (40) in the proof of Lemma 16, we have

$$
\bar{y}_{t+1,i} = y_{t,i} - \frac{\delta}{16L}\sqrt{\frac{\rho\epsilon}{n}} \cdot \hat{g}_{t,i},
$$

and

$$
|\bar{y}_{t+1,i}| \leq \left|y_{t,i} - \frac{\delta}{16L}\sqrt{\frac{\rho\epsilon}{n}} \cdot \frac{g_{t,i}}{\|\mathbf{g}_t\|}\right| - \frac{\delta}{16L}\sqrt{\frac{\rho\epsilon}{n}}\left|\hat{g}_{t,i} - \frac{g_{t,i}}{\|\mathbf{g}_t\|}\right|, \tag{42}
$$

where the second term on the RHS of (42) satisfies

$$
\frac{\delta}{16L}\sqrt{\frac{\rho\epsilon}{n}}\left|\hat{g}_{t,i} - \frac{g_{t,i}}{\|\mathbf{g}_t\|}\right| \leq \frac{\delta}{16L}\sqrt{\frac{\rho\epsilon}{n}}\left\|\hat{\mathbf{g}}_t - \frac{\mathbf{g}_t}{\|\mathbf{g}_t\|}\right\| \leq \frac{\delta\hat{\delta}}{16L}\sqrt{\frac{\rho\epsilon}{n}} \leq \frac{\delta(\rho\epsilon)^{1/4}}{128\mathscr{T}n}\sqrt{\frac{\pi}{L}}
$$

by Lemma 14. Moreover, the first term on the RHS of (42) satisfies

$$
y_{t,i} - \frac{\delta}{16L}\sqrt{\frac{\rho\epsilon}{n}} \cdot \frac{g_{t,i}}{\|\mathbf{g}_t\|} = y_{t,i} - \frac{\delta}{16L}\sqrt{\frac{\rho\epsilon}{n}} \cdot \frac{\mathbf{u}_i^\top \nabla^2 f(\mathbf{0})\mathbf{u}_i y_{t,i}}{\|\mathbf{g}_t\|} \leq \left(1 + \frac{\delta\rho\epsilon}{32\|\mathbf{g}_t\|L\sqrt{n}}\right)y_{t,i},
$$

Consequently, we have

$$
|\bar{y}_{t+1,i}| \leq \left(1 + \frac{\delta\rho\epsilon}{32\|\mathbf{g}_t\|L\sqrt{n}}\right)|y_{t,i}| + \frac{\delta(\rho\epsilon)^{1/4}}{128\mathscr{T}n}\sqrt{\frac{\pi}{L}}.
$$

Meanwhile,

$$
\begin{aligned}
|\bar{y}_{t+1,1}| &\geq \left|y_{t,1} - \frac{\delta}{16L}\sqrt{\frac{\rho\epsilon}{n}} \cdot \frac{g_{t,1}}{\|\mathbf{g}_t\|}\right| - \frac{\delta}{16L}\sqrt{\frac{\rho\epsilon}{n}}\left|\hat{g}_{t,1} - \frac{g_{t,1}}{\|\mathbf{g}_t\|}\right| \\
&\geq \left(1 + \frac{\delta\rho\epsilon}{16\|\mathbf{g}_t\|L\sqrt{n}}\right)|y_{t,1}| - \frac{\delta(\rho\epsilon)^{1/4}}{128\mathscr{T}n}\sqrt{\frac{\pi}{L}} \\
&\geq \left(1 + \frac{\delta\rho\epsilon}{24\|\mathbf{g}_t\|L\sqrt{n}}\right)|y_{t,1}|,
\end{aligned}
$$

where the last inequality is due to the fact that $|y_{t,1}| \geq \frac{\delta}{8}\sqrt{\frac{\pi}{n}}$ by Lemma 16. Hence, for any $t \in [\mathscr{T}-1]$ we have

$$
\begin{aligned}
\frac{|y_{t+1,i}|}{|y_{t+1,1}|} &= \frac{|\bar{y}_{t+1,i}|}{|\bar{y}_{t+1,1}|} \\
&\leq \frac{\left(1 + \frac{\delta\rho\epsilon}{32\|\mathbf{g}_t\|L\sqrt{n}}\right)|y_{t,i}| + \frac{\delta(\rho\epsilon)^{1/4}}{128\mathscr{T}n}\sqrt{\frac{\pi}{L}}}{\left(1 + \frac{\delta\rho\epsilon}{24\|\mathbf{g}_t\|L\sqrt{n}}\right)|y_{t,1}|} \\
&\leq \frac{\left(1 + \frac{\delta\rho\epsilon}{32\|\mathbf{g}_t\|L\sqrt{n}}\right)|y_{t,i}|}{\left(1 + \frac{\delta\rho\epsilon}{24\|\mathbf{g}_t\|L\sqrt{n}}\right)|y_{t,1}|} + \frac{(\rho\epsilon)^{1/4}}{8\mathscr{T}\sqrt{nL}} \\
&\leq \left(1 - \frac{\delta\rho\epsilon}{192\|\mathbf{g}_t\|L\sqrt{n}}\right)\frac{|y_{t,i}|}{|y_{t,1}|} + \frac{(\rho\epsilon)^{1/4}}{8\mathscr{T}\sqrt{nL}}.
\end{aligned}
$$

Since $f$ is $L$-smooth, we have

$$
\|\mathbf{g}_t\| \leq +L\|\mathbf{y}_t\| \leq L,
$$

which leads to

$$
\begin{aligned}
\frac{|y_{t+1,i}|}{|y_{t+1,1}|} &\leq \left(1 - \frac{\delta\rho\epsilon}{192\|\mathbf{g}_t\|L\sqrt{n}}\right)\frac{|y_{t,i}|}{|y_{t,1}|} + \frac{(\rho\epsilon)^{1/4}}{8\mathscr{T}\sqrt{nL}} \\
&\leq \left(1 - \frac{\delta\rho\epsilon}{192L^2\sqrt{n}}\right)\frac{|y_{t,i}|}{|y_{t,1}|} + \frac{(\rho\epsilon)^{1/4}}{8\mathscr{T}\sqrt{nL}}.
\end{aligned}
$$

Thus,

$$
\begin{aligned}
\frac{|y_{\mathscr{T},i}|}{|y_{\mathscr{T},1}|} &\leq \left(1 - \frac{\delta\rho\epsilon}{192L^2\sqrt{n}}\right)^{\mathscr{T}}\frac{|y_{0,i}|}{|y_{0,1}|} + \sum_{t=1}^{\mathscr{T}}\frac{(\rho\epsilon)^{1/4}}{6\mathscr{T}\sqrt{nL}}\left(1 - \frac{\delta\rho\epsilon}{192L^2\sqrt{n}}\right)^{\mathscr{T}-t} \\
&\leq \left(1 - \frac{\delta\rho\epsilon}{192L^2\sqrt{n}}\right)^{\mathscr{T}}\frac{|y_{0,i}|}{|y_{0,1}|} + \frac{(\rho\epsilon)^{1/4}}{8\sqrt{nL}} \leq \frac{(\rho\epsilon)^{1/4}}{4\sqrt{nL}}.
\end{aligned}
$$

$\square$

Equipped with Lemma 17, we are now ready to prove Lemma 15.

*Proof of Lemma 15.* We consider the case where $|y_{0,1}| \geq \frac{\delta}{2}\sqrt{\frac{\pi}{n}}$, which happens with probability

$$
\Pr\left\{|y_{0,1}| \geq \frac{\delta}{2}\sqrt{\frac{\pi}{n}}\right\} \geq 1 - \frac{\delta}{2}\sqrt{\frac{\pi}{n}} \cdot \frac{\text{Vol}(\mathcal{S}^{n-2})}{\text{Vol}(\mathcal{S}^{n-1})} \geq 1 - \delta.
$$

In this case, by Lemma 17 we have

$$
|y_{\mathscr{T},1}|^2 = \frac{|y_{\mathscr{T},1}|^2}{\sum_{i=1}^{n}|y_{\mathscr{T},i}|^2} = \left(1 + \sum_{i=2}^{n}\left(\frac{|y_{\mathscr{T},i}|}{|y_{\mathscr{T},1}|}\right)^2\right)^{-1} \geq \left(1 + \frac{\sqrt{\rho\epsilon}}{16L}\right)^{-1} \geq 1 - \frac{\sqrt{\rho\epsilon}}{8L},
$$

and

$$
\|\mathbf{y}_{\mathscr{T},\perp}\|^2 = 1 - |y_{\mathscr{T},1}|^2 \leq \frac{\sqrt{\rho\epsilon}}{8L}.
$$

Let $s$ be the smallest integer such that $\hat{\lambda}_s \geq 0$. Then the output $\hat{\mathbf{e}} = \mathbf{y}_{\mathscr{T}}$ of Algorithm 9 satisfies

$$
\begin{aligned}
\hat{\mathbf{e}}^\top \nabla^2 f(\mathbf{x})\hat{\mathbf{e}} &= |y_{\mathscr{T},1}|^2 \mathbf{u}_1^\top \nabla^2 f(\mathbf{x})\mathbf{u}_1 + \mathbf{y}_{\mathscr{T},\perp}^\top \nabla^2 f(\mathbf{x})\mathbf{y}_{\mathscr{T},\perp} \\
&\leq -\sqrt{\rho\epsilon} \cdot |y_{\mathscr{T},1}|^2 + L\sum_{i=s}^{d}\|\mathbf{y}_{\mathscr{T},i}\|^2 \\
&\leq -\sqrt{\rho\epsilon} \cdot |y_{\mathscr{T},1}|^2 + L\|\mathbf{y}_{\mathscr{T},\perp}\|^2 \leq -\frac{\sqrt{\rho\epsilon}}{4}.
\end{aligned}
$$

The query complexity of Algorithm 9 only comes from the Hessian-vector product estimation step in Line 4, which equals

$$\mathscr{T} \cdot O\big(n \log\big(n\rho L^2/\gamma_{\mathbf{x}}\gamma_{\mathbf{y}}^2\epsilon\hat{\delta}^2\big)\big) = O\left(\frac{L^2 n^{3/2}}{\delta\rho\epsilon} \log^2 \frac{nL}{\delta\sqrt{\rho\epsilon}}\right).$$

$\square$

## C.3 PROOF OF LEMMA 4

*Proof.* By Lemma 10 and Lemma 15, at least one of the two unit vectors $\mathbf{v}_1, \mathbf{v}_2$ is a negative curvature direction. Quantitatively, with probability at least $1 - \delta$, at least one of the following two inequalities is true:

$$\mathbf{v}_1^\top \nabla^2 f(\mathbf{z})\mathbf{v}_1 \leq -\frac{\sqrt{\rho\epsilon}}{4}, \qquad \mathbf{v}_2^\top \nabla^2 f(\mathbf{z})\mathbf{v}_2 \leq -\frac{\sqrt{\rho\epsilon}}{4}.$$

WLOG we assume the first inequality is true. Denote $\eta = \frac{1}{2}\sqrt{\frac{\epsilon}{\rho}}$. Given that $f$ is $\rho$-Hessian Lipschitz, we have

$$f(\mathbf{z}_{1,+}) \leq f(\mathbf{z}) + \eta\langle\nabla f(\mathbf{z}), \mathbf{v}_1\rangle + \int_0^\eta \left(\int_0^a \left(-\frac{\sqrt{\rho\epsilon}}{4} + \rho b\right) \mathrm{d}b\right) \mathrm{d}a$$

$$= f(\mathbf{z}) + \eta\langle\nabla f(\mathbf{z}), \mathbf{v}_1\rangle - \frac{1}{48}\sqrt{\frac{\epsilon^3}{\rho}},$$

and

$$f(\mathbf{z}_{1,-}) \leq f(\mathbf{z}) - \eta\langle\nabla f(\mathbf{z}), \mathbf{v}_1\rangle + \int_0^\eta \left(\int_0^a \left(-\frac{\sqrt{\rho\epsilon}}{4} + \rho b\right) \mathrm{d}b\right) \mathrm{d}a$$

$$= f(\mathbf{z}) - \eta\langle\nabla f(\mathbf{z}), \mathbf{v}_1\rangle - \frac{1}{48}\sqrt{\frac{\epsilon^3}{\rho}}.$$

Hence,

$$\frac{f(\mathbf{z}_{1,+}) + f(\mathbf{z}_{1,-})}{2} \leq f(\mathbf{z}) - \frac{1}{48}\sqrt{\frac{\epsilon^3}{\rho}},$$

which leads to

$$f(\mathbf{z}_{\text{out}}) \leq \min\{f(\mathbf{z}_{1,+}), f(\mathbf{z}_{1,-})\} \leq f(\mathbf{z}) - \frac{1}{48}\sqrt{\frac{\epsilon^3}{\rho}}.$$

By Lemma 10 and Lemma 15, the query complexity of Algorithm 6 equals

$$O\left(\frac{L^2 n^{3/2}}{\delta\rho\epsilon} \log^2 \frac{nL}{\delta\sqrt{\rho\epsilon}}\right).$$

$\square$

## C.4 ESCAPE FROM SADDLE POINT VIA NEGATIVE CURVATURE FINDING

**Lemma 18.** *In the setting of Problem 4, if the iterations $\mathbf{x}_{s,0}, \ldots, \mathbf{x}_{s,\mathscr{T}}$ of Algorithm 5 satisfy*

$$f(\mathbf{x}_{s,\mathscr{T}}) - f(\mathbf{x}_{s,0}) \geq -\frac{1}{48}\sqrt{\frac{\epsilon^3}{\rho}},$$

*then the number of $\epsilon$-FOSP among $\mathbf{x}_{s,0}, \ldots, \mathbf{x}_{s,\mathscr{T}}$ is at least $\mathscr{T} - \frac{3L}{32\sqrt{\rho\epsilon}}$.*

*Proof.* For any iteration $t \in [\mathscr{T}]$ with $\|\nabla f(\mathbf{x}_{s,t})\| > \epsilon$, by Theorem 1 we have

$$\left\| \hat{\mathbf{g}}_t - \frac{\nabla f(\mathbf{x}_{s,t})}{\|\nabla f(\mathbf{x}_{s,t})\|} \right\| \le \delta = \frac{1}{6},$$

indicating

$$f(\mathbf{x}_{s,t+1}) - f(\mathbf{x}_{s,t}) \le f(\mathbf{y}_{s,t}) - f(\mathbf{x}_{s,t})$$

$$\le \langle \nabla f(\mathbf{x}_{s,t}), \mathbf{x}_{s,t+1} - \mathbf{x}_{s,t} \rangle + \frac{L}{2} \|\mathbf{x}_{s,t+1} - \mathbf{x}_{s,t}\|^2$$

$$\le -\frac{\epsilon}{3L} \langle \nabla f(\mathbf{x}_{s,t}), \hat{\mathbf{g}}_t \rangle + \frac{L}{2} \left( \frac{\epsilon}{3L} \right)^2$$

$$\le -\frac{\epsilon}{3L} \|\nabla f(\mathbf{x}_{s,t})\|(1-\delta) + \frac{\epsilon^2}{18L} \le -\frac{2\epsilon^2}{9L}.$$

That is to say, for any $t \in [\mathscr{T}]$ such that $\mathbf{x}_{s,t}$ is not an $\epsilon$-FOSP, the function value will decrease at least $\frac{2\epsilon^2}{9L}$ in this iteration. Moreover, given that

$$f(\mathbf{x}_{s,t+1}) = \min\{f(\mathbf{x}_{s,t}), f(\mathbf{y}_{s,t})\} \le f(\mathbf{x}_{s,t})$$

and

$$f(\mathbf{x}_{s,0}) - f(\mathbf{x}_{s,\mathscr{T}}) \le \frac{1}{48} \sqrt{\frac{\epsilon^3}{\rho}},$$

we can conclude that the number of $\epsilon$-FOSP among $\mathbf{x}_{s,1}, \ldots, \mathbf{x}_{s,\mathscr{T}}$ is at least

$$\mathscr{T} - \frac{1}{48} \sqrt{\frac{\epsilon^3}{\rho}} \cdot \frac{9L}{2\epsilon^2} = \mathscr{T} - \frac{3L}{32\sqrt{\rho\epsilon}}.$$

$\square$

**Lemma 19.** *In the setting of Problem 4, if there are less than $\frac{8\mathscr{T}}{9}$ $\epsilon$-SOSP among the iterations $\mathbf{x}_{s,0}, \ldots, \mathbf{x}_{s,\mathscr{T}}$ of Algorithm 5, with probability at least $1 - (1 - p(1-\delta))^{\mathscr{T}/18}$ we have*

$$f(\mathbf{x}_{s+1,0}) - f(\mathbf{x}_{s,0}) \le -\frac{1}{48} \sqrt{\frac{\epsilon^3}{\rho}}.$$

*Proof.* If $f(\mathbf{x}_{s,\mathscr{T}}) - f(\mathbf{x}_{s,0}) \le -\frac{1}{48} \sqrt{\frac{\epsilon^3}{\rho}}$, we directly have

$$f(\mathbf{x}_{s+1,0}) - f(\mathbf{x}_{s,0}) = \min\{f(\mathbf{x}_{s,0}), \ldots, f(\mathbf{x}_{s,\mathscr{T}}), f(\mathbf{x}'_{s,0}), \ldots, f(\mathbf{x}'_{s,\mathscr{T}})\} - f(\mathbf{x}_{s,0})$$

$$\le f(\mathbf{x}_{s,\mathscr{T}}) - f(\mathbf{x}_{s,0}) \le -\frac{1}{48} \sqrt{\frac{\epsilon^3}{\rho}}.$$

Hence, we only need to prove the case with $f(\mathbf{x}_{s+1,0}) - f(\mathbf{x}_{s,0}) > -\frac{1}{48} \sqrt{\frac{\epsilon^3}{\rho}}$, where by Lemma 18 the number of $\epsilon$-FOSP among $\mathbf{x}_{s,0}, \ldots, \mathbf{x}_{s,\mathscr{T}}$ is at least $\mathscr{T} - \frac{3L}{32\sqrt{\rho\epsilon}}$. Since there are less than $\frac{8\mathscr{T}}{9}$ $\epsilon$-SOSP among the iterations $\mathbf{x}_{s,0}, \ldots, \mathbf{x}_{s,\mathscr{T}}$, there exists

$$\mathscr{T} - \frac{3L}{32\sqrt{\rho\epsilon}} - \frac{8\mathscr{T}}{9} \ge \frac{\mathscr{T}}{18}$$

different values of $t \in [\mathscr{T}]$ such that

$$\|\nabla f(\mathbf{x}_{s,t})\| \le \epsilon, \quad \lambda_{\min}(\nabla^2 f(\mathbf{x})) \le -\sqrt{\rho\epsilon}.$$

For each such $t$, with probability $p$ the subroutine `Comparison-NCD` (Algorithm 6) is executed in this iteration. Conditioned on that, with probability at least $1 - \delta$ its output $\mathbf{x}'_{s,t}$ satisfies

$$f(\mathbf{x}'_{s,t}) - f(\mathbf{x}_{s,t}) \le -\frac{1}{48} \sqrt{\frac{\epsilon^3}{\rho}}$$

by Lemma 4. Hence, with probability at least

$$1 - (1 - p(1 - \delta))^{\mathscr{T}/18},$$

there exists a $t' \in [\mathscr{T}]$ with

$$f(\mathbf{x}'_{s,t'}) - f(\mathbf{x}_{s,t'}) \leq -\frac{1}{48}\sqrt{\frac{\epsilon^3}{\rho}},$$

which leads to

$$f(\mathbf{x}_{s+1,0}) - f(\mathbf{x}_{s,0}) = \min\{f(\mathbf{x}_{s,0}), \ldots, f(\mathbf{x}_{s,\mathscr{T}}), f(\mathbf{x}'_{s,0}), \ldots, f(\mathbf{x}'_{s,\mathscr{T}})\} - f(\mathbf{x}_{s,0})$$

$$\leq f(\mathbf{x}'_{s,t'}) - f(\mathbf{x}_{s,t'}) \leq -\frac{1}{48}\sqrt{\frac{\epsilon^3}{\rho}},$$

where the second inequality is due to the fact that $f(\mathbf{x}_{s,t'}) \leq (\mathbf{x}_{s,0})$ for any possible value of $t'$ in $[\mathscr{T}]$. $\qquad\square$

*Proof of Theorem 5.* We assume for any $s = 1, \ldots, \mathcal{S}$ with $\mathbf{x}_{s,0}, \ldots, \mathbf{x}_{s,\mathscr{T}}$ containing less than $\frac{8\mathscr{T}}{9}$ $\epsilon$-SOSP we have

$$f(\mathbf{x}_{s+1,0}) - f(\mathbf{x}_{s,0}) \leq -\frac{1}{48}\sqrt{\frac{\epsilon^3}{\rho}}.$$

Given that there are at most $\mathcal{S}$ different values of $s$, by Lemma 19, the probability of this assumption being true is at least

$$\left(1 - (1 - p(1 - \delta))^{\mathscr{T}/18}\right)^{\mathcal{S}} \geq \frac{8}{9}. \tag{43}$$

Moreover, given that

$$\sum_{s=1}^{\mathcal{S}} f(\mathbf{x}_{s+1,0}) - f(\mathbf{x}_{s,0}) = f(\mathbf{x}_{\mathcal{S}+1,0}) - f(\mathbf{0}) \geq f^* - f(0) \geq -\Delta$$

there are at least $\frac{27}{32}\mathcal{S}$ different values of $s = 1, \ldots, \mathcal{S}$ with

$$f(\mathbf{x}_{s+1,0}) - f(\mathbf{x}_{s,0}) \leq -\frac{1}{48}\sqrt{\frac{\epsilon^3}{\rho}},$$

as we have $f(\mathbf{x}_{s+1,0}) \leq f(\mathbf{x}_{s,0})$ for any $s$. Hence, in this case the proportion of $\epsilon$-SOSP among all the iterations is at least

$$\frac{\frac{27}{32}\mathcal{S} \cdot \frac{8}{9}\mathscr{T}}{\mathcal{S}\mathscr{T}} = \frac{3}{4}.$$

Combined with (43), the overall success probability of outputting an $\epsilon$-SOSP is at least $\frac{3}{4} \times \frac{8}{9} = \frac{2}{3}$.

The query complexity of Algorithm 5 comes from both the gradient estimation step in Line 5 and the negative curvature descent step in Line 8. By Theorem 1, the query complexity of the first part equals

$$\mathcal{S}\mathscr{T} \cdot O(n\log(n/\delta)) = O\left(\frac{\Delta L^2 n^{3/2}}{\rho^{1/2}\epsilon^{5/2}}\log n\right),$$

whereas the expected query complexity of the second part equals

$$\mathcal{S}\mathscr{T}p \cdot O\left(\frac{L^2 n^{3/2}}{\delta\rho\epsilon}\log^2\frac{nL}{\delta\sqrt{\rho\epsilon}}\right) = O\left(\frac{\Delta L^2 n^{3/2}}{\rho^{1/2}\epsilon^{5/2}}\log^3\frac{nL}{\sqrt{\rho\epsilon}}\right).$$

Hence, the overall query complexity of Algorithm 5 equals

$$O\left(\frac{\Delta L^2 n^{3/2}}{\rho^{1/2}\epsilon^{5/2}}\log^3\frac{nL}{\sqrt{\rho\epsilon}}\right).$$

$\qquad\square$