# OpenReview forum: "Comparisons Are All You Need for Optimizing Smooth Functions"
_ICLR.cc/2025/Conference — Submitted to ICLR 2025_

### Official Review · Reviewer_Z643 · 2024-10-26

**Soundness:** 3
**Presentation:** 3
**Contribution:** 1
**Rating:** 3
**Confidence:** 4

**Summary:**

The paper proposes algorithms for optimization where the learner only has access to comparisons between the function values at two points. Under this setting, the authors propose algorithms for both convex and non-convex settings that achieve order-optimal performance in terms of sample complexity.

**Strengths:**

The proposed algorithms achieve optimal performance across a variety of cases, which cover the most commonly encountered scenarios. This provides a holistic view of comparison oracle based optimization.

I think the design of the routine for estimating if the point has negative curvature is also interesting and novel.

**Weaknesses:**

I think the main weakness of the paper is limited novelty. As pointed by the authors, the key technical contribution is designing the routine comparison-GDE, which builds an estimate of the gradient at a given point by querying points in the vicinity. After that all the algorithms, their design and analysis largely from existing literature.

As far as I understand, this comparison-GDE is also very similar to the gradient estimation technique in dueling convex optimization (Saha et al, 2021, 2022). The only difference is that the authors in Saha et al. use a direction drawn uniformly at random to estimate the gradient direction while the approach in this work relies on choosing the directions deterministically. In terms of both analysis and algorithm design, I see little different between the two approaches. Even in terms of computational costs, the difference is negligible.

As mentioned in the previous response, I think the curvature estimation routine is new and interesting, but is not sufficient on its own to warrant a publication.

On a separate note, I think the motivation for the problem is also somewhat misplaced. Learning via preference based feedback for RLHF stems from a practical concern of challenges in reward estimation. For optimization, this is not the case because function is well-defined in a lot of cases and from a ML optimization point of view, comparing two function values is not necessarily simpler than computing the gradient on a batch of samples.

**Questions:**

1. As discussed above, what is the difference between the proposed approach and the one in Saha et al, 2021. I am more curious about the difference at a fundamental level, not the superficial difference in terms of random vectors vs deterministic choice.

2. Also, it seems like you separate comparison oracle based optimization from dueling convex optimization. To the best of my understanding, these are technically the same thing. If anything, dueling optimization is slightly more general, allowing for noisy observations.

3. For the case where your $x$ is very close to the boundary, you will have to slightly modify the algorithm to ensure that the query points are within the domain. It is relatively simple to fix, but something that needs to be kept in mind.

4. Clearly, there is an inherent assumption that the observations are noiseless. Can the algorithm be modified to adapt to the scenario when the preference feedback is noisy?

---

> ### Author Response · Authors · 2024-11-24
>
> We appreciate your great effort and valuable suggestions.
>
> Regarding your question on the originality and the technical contribution of our paper, we agree that you raised a good point which has also been pointed out by other reviewers. Please refer to the ''novelty and technical contribution'' section  general response for a detailed discussion.
>
> More specifically, your question about the differences between our gradient direction estimation algorithm and the approach in Section 4 of [1], we note that while both methods achieve a similar computational cost, their underlying mechanisms are fundamentally distinct. Specifically, the method in [1] leverages the statistical properties of random vectors, making it a randomized approach that operates in a single round. In contrast, our method deterministically computes the angles between pairs of gradient components, at the cost of requiring multiple rounds. We would be super curious if there is a way connecting these two approaches. If you have any insights or suggestions on this, we would be delighted to hear them and would more than happy to incorporate additional discussion into our final version.
>
> Regarding your comment on comparison-based optimization and the dueling convex optimization setting, we agree that they are basically the same, with dueling optimization slightly more general as it accommodates stochastic information. However, we note that prior literature uses varying terminologies for similar concepts, including dueling optimization [1], comparison-based optimization [2], direct search method [3], and sub-zeroth-order optimization [4]. In the final version, we will expand our discussion to clarify the relations between these terminologies.
>
> Regarding your comment on the case where the query point $x$ is very close to the boundary, we thank you for pointing this out and will fix this in our final version.
>
> Regarding your question on the robustness of our algorithm to noise, we find this to be a very interesting and natural direction for future research. In the final version, we will update our open problems section to explicitly include this question.
>
> **References.**
>
> [1] Aadirupa Saha, Tomer Koren, and Yishay Mansour. Dueling convex optimization, 2021.
>
> [2] Yuanhao Wang, Qinghua Liu, and Chi Jin. Is RLHF more difficult than standard RL? a theoretical perspective, 2023.
>
> [3] El Houcine Bergou, Eduard Gorbunov, and Peter Richtárik. Stochastic three points method for unconstrained smooth minimization, 2020.
>
> [4] Mustafa O. Karabag, Cyrus Neary, and Ufuk Topcu. Smooth convex optimization using sub-zeroth-order oracles, 2021.

---

### Official Review · Reviewer_mSDt · 2024-11-04

**Soundness:** 3
**Presentation:** 3
**Contribution:** 2
**Rating:** 5
**Confidence:** 4

**Summary:**

The paper addresses optimization for scenarios where gradient computation is difficult or impossible. The authors focus on comparison-based methods, which leverage only the relative magnitudes of function values at different points, rather than their exact values or gradients. This approach is particularly relevant for cases like reinforcement learning from human feedback (RLHF) in training large language models, where only preferences, not precise rewards, are available.
The key technical contribution is a gradient direction estimation algorithm using only comparisons, which underlies the proposed optimization methods. Their work advances comparison-based optimization, demonstrating that comparisons can achieve similar performance to zeroth-order methods that require function evaluations. Future directions include adapting accelerated gradient methods to comparison-based frameworks and extending the applicability of these methods to machine learning, particularly in preference-based RL contexts.

**Strengths:**

Originality:
    The paper tackles an emerging topic by focusing on optimization using only comparisons, which has relevance in preference-based reinforcement learning (RL) and derivative-free optimization scenarios. This direction is increasingly important, especially with applications in reinforcement learning from human feedback (RLHF) for large language models.

Theoretical Quality:
    The paper provides rigorous theoretical complexity bounds for both convex and non-convex optimization, closely matching traditional zeroth-order methods. These analyses are well-supported mathematically, showing an in-depth understanding of comparison-based optimization.

Clarity:
    The methods and algorithms are clearly explained with concise mathematical descriptions, which makes the theoretical contributions accessible and reasonably easy to follow. Definitions, assumptions, and theorems are presented in an organized manner, contributing to overall readability.

**Weaknesses:**

Originality:
    While the topic is relevant, the contributions primarily extend existing methods in comparison-based and zeroth-order optimization rather than introduce fundamentally new concepts. The work’s novelty is incremental, repurposing established techniques like normalized gradient descent within a comparison-based framework.

Lack of Empirical Validation:
    The paper is exclusively theoretical, lacking experiments or simulations to verify that the proposed methods work effectively in practice. This absence limits the reliability of the results, as real-world performance, especially under noisy conditions, may deviate from the theoretical predictions.

Contextualization:
    The paper’s literature review and discussion sections do not fully differentiate the contributions from prior work, particularly in terms of practical advantages over other comparison-based methods. This leaves the impression that the paper does not convincingly address why its approach is preferable in applications.

Significance:
    The impact of the contributions is limited due to the lack of application-driven validation. While the theoretical results are valuable, without empirical demonstrations or clear real-world applicability, the significance for the broader ML community remains uncertain.

**Questions:**

1. Including experiments would significantly strengthen the paper. Can the authors provide empirical results or simulations to validate the theoretical claims, especially in noisy or real-world conditions?

2. Given the motivation in reinforcement learning from human feedback (RLHF), how might the proposed algorithms be adapted or tested in RL scenarios? A discussion on how these methods could integrate with or improve current preference-based RL approaches would add value.

---

> ### Author Response · Authors · 2024-11-24
>
> We appreciate your great effort and valuable suggestions.
>
> Regarding your question on the originality and the technical contribution of our paper, we agree that you raised a good point which has also been pointed out by other reviewers. Please refer to the ``novelty and technical contribution'' section  general response for a detailed discussion.
>
> Regarding your comment on the contextualization and comparison with prior works, we agree that you raised a good point which has also been pointed out by other reviewers. Please refer to the ``presentation and comparison with previous works'' section in our general response for a detailed discussion.
>
> Regarding your comment on the significance of our work as well as lack of empirical validation, we agree that you raised a good point which has also been pointed out by other reviewers. Please refer to the general response for a detailed discussion.

---

### Official Review · Reviewer_TP2H · 2024-11-04

**Soundness:** 3
**Presentation:** 2
**Contribution:** 3
**Rating:** 5
**Confidence:** 3

**Summary:**

The paper studies comparison-based optimization algorithms and prove the convergence.

**Strengths:**

The paper propose several comparison-based optimization algorithms and show the convergence to first-order and second-order optimal solutions. Both convex and non-convex scenarios are analyzed.

**Weaknesses:**

1. I would suggest the authors provide a table of sample complexity of the algorithms for comparison with the results in the literature. Although the authors carefully discuss the existing results, the query complexity is still not very clear.
2. The writing should be improved. First, interpretation of each theoretical result is necessary. I would suggest highlighting the technical contributions for each theorem and lemma. Meanwhile, be careful about the equation numbering. (2) and (3) are missing in the main text; see proof of Theorem 1.

**Questions:**

As comparison-based optimization algorithms can be used in RLHF and LLMs, I am curious about the practical performance of these algorithms over real-world dataset and models.

---

> ### Author Response · Authors · 2024-11-24
>
> We appreciate your great effort and valuable suggestions.
>
> Regarding on your question on the comparison of our results with prior works and your comment on providing a table of sample complexity, we agree that you raised a good point which has also been pointed out by other reviewers. Please refer to the general response for a detailed discussion.
>
> Regarding your comment on the writing, we agree with you that highlighting the technical contributions for each theorem and lemma would greatly benefit the readability of the paper. Moreover, we thank you for pointing out the issue in equation numbering. We will fix these issues in our final version.
>
> Regarding your question on the practical performance of our algorithms and possible applications in RLHF and LLMs, we agree that you raised a good point which has also been pointed out by other reviewers. Please refer to the ''Motivation and possible applications in practical scenarios'' section in our general response for a detailed discussion.

---

> > ### Comment · Reviewer_TP2H · 2024-11-27
> >
> > Thank you for the response. I believe my evaluation is fair.

---

### Official Review · Reviewer_PHeS · 2024-11-04

**Soundness:** 3
**Presentation:** 3
**Contribution:** 3
**Rating:** 6
**Confidence:** 3

**Summary:**

The authors consider the gradient-free method to optimize functions without requiring function value evaluations, relying only on comparisons between function values. In particular, for convex optimization problems, an $\epsilon$-optimal solution can be found, using Theorem 2, in $\mathcal{O}(d/\epsilon)$ oracle queries, which matches the result of [Nesterov & Spokoiny, 2017] in terms of its dependence upon dimension $d$, at the same time, being worse in terms of its dependene upon $\epsilon$. The authors also suggest an algorithm with poly-logarithmic complexity based on cutting plane method. The key technical contribution is Theorem 1, that allows to construct an estimator of the direction of gradient.

**Strengths:**

The main text of the paper is well-written, and the main results of the submission are well-explained. The authors have put efforts to explain their technical contributions, in particular, the way how the estimates of normalized gradients of theorem 1 are propagated through the further results.

**Weaknesses:**

I find the novelty of the paper somehow limited, since the main results of the paper (Th. 2 and Th. 4) directly corresponds to the known results for Stochastic Three Points method (STP, [Bergou et al., 2020]), with the same oracle complexities. In this sense the main novelty of the submission comes from the fact that the method in the paper is purely comparison-based. Moreover, the sample complexity results achieved by, for example, Theorem 2, is worse in terms of its dependence upon tolerance level $\epsilon$, compared to the best known results, for example, [Nesterov & Spokoiny, 2017]. In this case I would be curious to find a setting, where the comparison-based method is more valuable, than STP.

**Questions:**

Is it possible to generalize the results presented in the paper for variational inequalities? In particular, is it possible to apply the results for the Nash equilibrium seeking problems?

---

> ### Author Response · Authors · 2024-11-24
>
> Thank you for your positive feedback and detailed suggestions!
>
> Regarding your question on the novelty and the technical contribution of our paper, we agree that you raised a good point which has also been pointed out by other reviewers. Please refer to the general response for a detailed discussion.
>
> Regarding generalizing our results to variational inequalities, this is an interesting question technically and worth future investigation. However, we are not very sure if that will provide better algorithms for finding $\epsilon$-optima of convex functions or finding $\epsilon$-SOSPs of nonconvex functions. Regarding applying our results for seeking Nash equilibria, that will require solving an minimax optimization problem, which has significant difference from solving a minimal/maximal optimization problem. More specifically, minimax optimization has form $\min_{x\in\mathcal{X}} \max_{y\in\mathcal{Y}} f(x,y)$, and we need to consider whether $f$ is convex or nonconvex in $x$ and also whether $f$ is concave or nonconcave in $y$. That gives different settings, and algorithms for convex-concave settings (potentially with or without strongly convex or strongly concave properties) [1] and nonconvex-nonconcave settings [2] are dramatically different. Considering this, minimax optimization constitutes an independent research direction over minimal optimization, and we also leave this for future work.
>
> **References.**
>
> [1] Tianyi Lin, Chi Jin, and Michael I. Jordan. Near-optimal algorithms for minimax optimization, 2020.
>
> [2] Jelena Diakonikolas, Constantinos Daskalakis, and Michael I. Jordan. Efficient Methods for Structured Nonconvex-Nonconcave Min-Max Optimization, 2021.

---

> > ### Comment · Reviewer_PHeS · 2024-11-29
> >
> > Dear authors,
> >
> > Thank you for the response, I will keep my current score.
> >
> > Best,
> > Reviewer PHeS

---

### Author Response · Authors · 2024-11-24
**General response**

Thank you to the reviewers for their thoughtful consideration of our paper. We appreciate your positive recognition of our contributions and helpful suggestions for improvement. Here we address some common questions that arose in multiple reviews. (More specific questions of individual reviewers are provided in each individual response.)

**Novelty and technical contribution.** Regarding the novelty and technical contribution of our work, we agree with reviewers that our techniques and their analysis share a fair amount of similarities with previous works on Stochastic Three Point method (STP) [1] and dueling convex optimization [2, 3]. However, our approach is fundamentally deterministic, more succinct, and therefore easier to analyze and extend. Notably, our results in Theorem 2 and Theorem 4 match the query complexities achieved in [1, 2, 3] using deterministic algorithms. Furthermore, we establish Theorem 3, which addresses the high-precision regime and achieves an $n^2$ speedup compared to the previous state-of-the-art result [4].

Moreover, while we agree with the reviewers that the techniques underlying the aforementioned results are not excessively complex, we would like to draw your attention to Theorem 5 in Section 4.2, which addresses escaping saddle points using comparison queries -- this constitutes the main technical contribution of our work. Specifically, we provide *the first provable guarantee for finding an $\epsilon$-SOSP of a nonconvex function using comparison-based methods*. The corresponding algorithms (Algorithm 5 and Algorithm 6) and their analysis are fundamentally different from existing approaches.

Previously, there was no clear intuition about whether finding an $\epsilon$-SOSP using only comparison queries was even feasible. On the other hand, among existing results using comparison-based queries, [1] can only find a first-order stationary point (FOSP) of a nonconvex smooth function, and [2, 3] only applies to convex functions. On the other hand, existing first-order methods [5, 6, 7] rely on a critical subroutine that approximates the Hessian-vector product by computing the difference of gradients at nearby points. This approach inherently depends on access to both the gradient's direction and its norm, making it incompatible with comparison-based methods.

To address this issue, we develop a novel subroutine Comparison-NCD (Algorithm 6) that approximates the direction of the Hessian-vector product. Although Comparison-NCD has a clear geometric intuition, its analysis is surprisingly difficult, given that the procedure of approximating a Hessian-vector product using gradient queries is already a delicate procedure that is vulnerable to errors, and estimating gradient directions using comparison queries is an intrinsically numerical instable task when the gradient has a small norm. Hence, we design and combine two specialized subroutines, each tailored to handle a distinct hard case. The error analysis for both subroutines is technically involved with contributions on its own.

**Presentation and Comparison with previous works.**
We appreciate the reviewers' suggestion for a more thorough discussion on the advantages of our approach compared to prior works. To address this, we will expand the literature review on page 2 in the final version to provide a clearer context for our contributions, and integrate some of the arguments from the above ''novelty and technical contribution'' bullet point. Additionally, we will include a table summarizing the complexities of our algorithms alongside those of prior works.

---

### Author Response · Authors · 2024-11-24
**General response continued**

**Motivation and possible applications in practical scenarios.**
Regarding your question about the motivation for our work, we agree that learning through preference-based feedback for RLHF arises from practical concerns. However, we also want to emphasize that this problem has been widely studied in RL theory since 2012 (see, for example, [8, 9]). From a theoretical perspective, optimization using comparison-based queries is a natural generalization of this setting. Furthermore, comparison-based optimization is of independent interest in optimization theory. Its ability to provide deeper insights into the underlying mathematical structure has led to the exploration of its variants within the context of direct search methods [10, 11].

Regarding the connection to LLMs, recent works such as [12] found out that LLMs whose parameters are taken from $\{-1, 0, 1\}$ match the full-precision (i.e., FP16 or BF16) Transformer LLM with the same model size and training tokens in terms of both perplexity and end-task performance. This suggests an interesting perspective for the optimization theory behind LLMs to be investigated in the future.

Given the purely theoretical nature of our work, it may be challenging to directly apply our methods to real-world scenarios and achieve advantages over existing approaches. Nevertheless, we hope our work can provide inspiration for future empirical studies. Additionally, we will provide numerical experiments in our final version to validate our theoretical claims.

**References.**

[1] El Houcine Bergou, Eduard Gorbunov, and Peter Richtárik. Stochastic three points method for unconstrained smooth minimization, 2020.

[2] Aadirupa Saha, Tomer Koren, and Yishay Mansour. Dueling convex optimization, 2021.

[3] Aadirupa Saha, Tomer Koren, and Yishay Mansour. Dueling convex optimization with general preferences, 2022.

[4] Mustafa O. Karabag, Cyrus Neary, and Ufuk Topcu. Smooth convex optimization using sub-zeroth-order oracles, 2021.

[5] Zeyuan Allen-Zhu and Yuanzhi Li. Neon2: Finding local minima via first-order oracles, 2018.

[6] Yi Xu, Rong Jin, and Tianbao Yang. First-order stochastic algorithms for escaping from saddle points in almost linear time, 2018.

[7] Chenyi Zhang and Tongyang Li. Escape saddle points by a simple gradient-descent based algorithm, 2021.

[8] Yuanhao Wang, Qinghua Liu, and Chi Jin. Is RLHF more difficult than standard RL? a theoretical perspective, 2023.

[9] Yisong Yue, Josef Broder, Robert Kleinberg, and Thorsten Joachims. The k-armed dueling bandits problem, 2012.

[10] Robert Michael Lewis, Virginia Torczon, and Michael W. Trosset. Direct search methods: then and now, 2000.

[11] Robert Hooke, and Terry A. Jeeves. Direct Search Solution of Numerical and Statistical Problems, 1961.

[12] Shuming Ma, Hongyu Wang, Lingxiao Ma, Lei Wang, Wenhui Wang, Shaohan Huang, Li Dong, Ruiping Wang, Jilong Xue, Furu Wei. The Era of 1-bit LLMs: All Large Language Models are in 1.58 Bits, 2024.

---

### Meta-Review · Area_Chair_DXh2 · 2024-12-23

**Metareview:**

This paper proposes new optimization methods without querying any gradient or even function value, but just querying comparisons---which one of the two queried points has a larger function value. This paper provides a comprehensive set of solutions for the newly proposed questions on convex, nonconvex (finding stationary point) and even escaping the saddle points. The dimension dependency of the proposed  methods are also sharp matching the best existing zero-th order methods. While this is an overall solid work, there are concerns on the technical novelty of this paper---while addressing a new problem, the techniques are rather similar to a line of prior works including Stochastic Three Point method (STP) and dueling convex optimization. There is also concerns regarding the practical justification of the newly proposed problem, while there is connection to RLHF, the connection is weak/loose and the authors were not able to convince the practical importance of the proposed problems. We recommend rejection at the current form of the paper, and suggest authors to address the aforementioned issues as well as revise title to more accurately reflect the content, and then try a future venue.

**Additional Comments On Reviewer Discussion:**

While the paper provides a solid set of results for newly proposed problem, the lack of technical novelty is the an important issue that is consistently raised by 3 out of the 4 reviewers. This is the major reason for rejection. Other reason such as lack of convincing motivation or empirical justification also contribute to the rejection.

---

### Decision · Program_Chairs · 2025-01-22

Reject